# The Green Era of Food Packaging: General Considerations and New Trends

**DOI:** 10.3390/polym14204257

**Published:** 2022-10-11

**Authors:** Enrico Maurizzi, Francesco Bigi, Andrea Quartieri, Riccardo De Leo, Luisa Antonella Volpelli, Andrea Pulvirenti

**Affiliations:** 1Department of Life Sciences, University of Modena and Reggio Emilia, 41125 Modena, Italy; 2Interdepartmental Research Centre for the Improvement of Agro-Food Biological Resources (BIOGEST-SITEIA), University of Modena and Reggio Emilia, 42124 Reggio Emilia, Italy

**Keywords:** biopolymers, antioxidant compounds, antimicrobial compounds, essential oils, nanoparticles

## Abstract

Recently, academic research and industries have gained awareness about the economic, environmental, and social impacts of conventional plastic packaging and its disposal. This consciousness has oriented efforts towards more sustainable materials such as biopolymers, paving the way for the “green era” of food packaging. This review provides a schematic overview about polymers and blends of them, which are emerging as promising alternatives to conventional plastics. Focus was dedicated to biopolymers from renewable sources and their applications to produce sustainable, active packaging with antimicrobial and antioxidant properties. In particular, the incorporation of plant extracts, food-waste derivatives, and nano-sized materials to produce bio-based active packaging with enhanced technical performances was investigated. According to recent studies, bio-based active packaging enriched with natural-based compounds has the potential to replace petroleum-derived materials. Based on molecular composition, the natural compounds can diversely interact with the native structure of the packaging materials, modulating their barriers, optical and mechanical performances, and conferring them antioxidant and antimicrobial properties. Overall, the recent academic findings could lead to a breakthrough in the field of food packaging, opening the gates to a new generation of packaging solutions which will be sustainable, customised, and green.

## 1. Introduction

Food technologies have played a crucial role since the beginning of human civilisation. Throughout history, the evolution of food processing and packaging has led to a constant increase of food quality and safety, improving the quality of human life [1]. Recently, human society has gained awareness about the impact of agri-food practices on our world, and these concerns have oriented the food sector towards the adoption of novel and sustainable technologies.

Among the main pillars of this multifaceted process, it is worth citing three lines of research that have deeply contributed to re-define the concept of “*Food Technology*” [2]: Substitution of thermal techniques and chemical sanitisation with green alternatives in order to reduce the consumption of resources and the impact on food quality.Extraction of added-value compounds from renewable sources (e.g., food by-products) and their application as alternatives to conventional preservatives and additives.Development of bio-based active packaging based on renewable biopolymers, aiming to reduce the use of petroleum-derived plastics in the food packaging sector, and to prolong the shelf-life of the products, preventing the generation of food waste.

This work provides a synthetic overview about the strategy trends which are leading the food-packaging industry towards green technology and sustainability criteria, reducing the energy consumption, waste generation, and footprints on the environment. A specific focus was dedicated to biodegradable polymers from renewable sources (e.g., agri-food by-products) and natural-derived compounds, and their application to produce active packaging items with antimicrobial, antioxidant, and nano-reinforced properties as prospective substitutes for conventional plastic materials. 

## 2. Bio-Based Packaging: General Considerations

Food packaging is a coordinated system aiming to preserve the safety and quality of the food products from their production to their end-use [3]. It plays a crucial role in human society as a fundamental component of the food supply chain [4].

Worldwide, it is estimated that one-third of produced food is disposed every year due to various factors including incorrect harvesting procedures, mechanical damage, and inadequate storage conditions, which result in microbial decay, oxidation, the degradation of nutrients, and loss of acceptability [5]. Therefore, the selection of adequate packaging solutions able to protect each targeted product and to maintain its quality is crucial to extend the food’s shelf life, thus preventing waste generation. 

Conventional packaging is commonly constituted by a one-time use item, immediately discarded after reaching the intermediate or final user [4]. Over a broad variety of materials, fossil-based plastics have dominated the food-packaging industry since their appearance during the Second World War [6] thanks to their enhanced barrier and mechanical properties, chemical resistance, durability, lightweight nature, availability, and cost-effectiveness [7].

Currently, the global production of plastics comprises about 320 million tons/year [8]. Data reveal that one-third of all produced plastic is dedicated to packaging materials [9]. Hence, the food-packaging industry is closely involved in the production of massive amounts of plastics, generating severe economic burdens and ecological impacts. 

The main concern of plastics is related to their non-sustainable nature since their source (petroleum) is not renewable (PE, PET, PP, etc.) [10,11,12]. Besides, single-use plastics are generally considered as not “environmentally friendly” due to their non-compostable nature and low recycling rate [13]. This ends up causing the accumulation of tremendous masses of waste in landfills and oceans, increasing wildlife mortality from ingestion and entanglement [14]. 

In the last few years, the awareness about the environmental impacts of plastic has grown both at personal and at community levels. On the one hand, consumers are increasingly demanding natural, high-quality foods, and food packaging that does not create pollution. On the other hand, governments are pushing towards the reduction of human impact on the environment. For example, the European Parliament focused its Sustainable Development Goals on the partial replacement of oil-based polymers with biodegradable polymers from renewable resources by 2030 (European Commission, 2015). This prompted researchers and companies to shift their efforts towards the exploration and exploitation of novel renewable resources and the development of sustainable packaging solutions, including films, coatings, and other items. 

Specifically, films are thin layers of material prepared through different technologies such as solution casting or extrusion as stand-alone structures. The prepared films are used to wrap the foods or to be placed between the layers of food products. Coatings are thin layers of material which are directly applied on the food surface, and act as a barrier between the external environment and the product during transportation, processing, and storage. Coatings are applied either by dipping the product in the coating solution or by directly spraying them over the product’s surface.

These novel packaging systems are designed to perform multiple functions. Along with the “classic” packaging activity, namely the interposition of a physical barrier between food and environment, they may operate as carriers of bioactive compounds with antioxidant, antimicrobial, or nutritional properties. These “active ingredients” aim to prolong the shelf life or increase the nutritional value of the packaged product [15]. Moreover, the addition of bioactive compounds can result in modified physicochemical, mechanical, and barrier properties since they chemically interact with the biopolymer structure. Hence, their wide application may allow improving or even adapting the functional features of packaging solutions for a broad variety of applications [16]. 

### 2.1. Compostable, Biodegradable, or Renewable?

Research and industries are pushing towards the usage of biodegradable polymers for food-packaging purposes. Additionally, the extensive exploitation of renewable resources has the potential to reduce the use of oil and other fuels. However, plastics produced by renewable resources are not necessarily compostable or biodegradable, and vice versa [17]. For example, cellulose, starch, and gelatin also maintain their biodegradability when obtained synthetically. Equally, when castor oil monomers are polymerised to produce Nylon 9, they lose their biodegradability [18]. In fact, biodegradation is correlated to the chemical structure of the compound rather that its origin. In this context, it is important to clearly state the definitions of biodegradation and compostability, allowing further introduction of the concept of biopolymers.

Biodegradation broadly defines an event in which a biomass is over 90% decomposed within 6 months via the action of enzymes and/or chemical degeneration associated with living organisms such as moulds, yeasts, and bacteria [(UNI EN 13432:2002)]. This process can be conducted both in aerobic and anaerobic conditions [19]. Other processes such as photodegradation, hydrolysis, and oxidation may also have an impact on the structure of biomass prior to or during biodegradation [20]. Compostability involves a series of processes (mainly conducted in industrial conditions) that exploit biodegradation to convert organic matter into the so-called “compost”, which must completely degrade in soil within 3 months by producing water, carbon dioxide, and other inorganic compounds [21].

In light of these statements, it is worth noting that the large-scale synthesis of compostable bioplastic using 100% renewable resources has not been realised yet. Until now, bioplastic usually comprises more than 50% (*w*/*w*) of renewable sources [18]. Several bioplastics include mixtures of synthetic compounds to improve the technical properties of the final product, extending its potential applications. Despite that, the current tendency is to replace synthetic additives with natural compounds with comparable functional properties and to enhance the use of biopolymers over fossil-derived materials to produce approximately 100% renewable and biodegradable plastics. 

### 2.2. Biopolymers 

According to the European Bioplastics association, biopolymers are defined as biodegradable, compostable, and biocompatible polymers derived from renewable resources [22]. They are broadly regarded as the most promising sustainable alternative to petrol-based synthetic polymers for food-packaging applications due to their compostable nature and film-forming ability [20]. 

Thanks to their technical variability, biopolymers are adaptable to various packaging technologies, offering a range of package products, including cups, covers, separation layers, and food containers. In particular, they can be used to prepare composite films and multi-layered coatings to prolong the shelf-life of food products. Moreover, biopolymers are compatible with functional ingredients including nutraceuticals, antioxidants, antimicrobials, probiotics, and additives [23]. 

Biopolymers have been classified into three categories according to their sources and synthesis: (I) polymers extracted from renewable biomasses, including polysaccharides, polypeptides, and lipids; (II) polymers synthetised from chemical polymerisation of bio-monomers (e.g., polylactic acid); and (III) polymers derived from microbial fermentation (e.g., polyhydroxy alkanoates) [19] (Figure 1). Besides, biopolymers can be distinguished according to their hydroplastic or thermoplastic behaviour [3]. 

Most biopolymers possess remarkable technical features for packaging applications due to their chemical complexity, as shown by the studies in Table 1. A brief description of the most common biopolymers is detailed in the following sub-sections. 

∙ Best performance of immobilisation at 2.5–3% of Na-alginate and CaCl_2_, with 400–600 mg/L of protease

#### 2.2.1. Polysaccharides

Polysaccharides are complex macromolecules consisting of repeated mono or disaccharide units linked via glycosidic bonds [43]. They are natural, easily accessible, non-toxic, and renewable.

Due to their complex structure, polysaccharides exhibit adequate mechanical resistance and high barrier to oxygen (O_2_) and carbon dioxide (CO_2_). The presence of hydroxyl groups lead to the formation of hydrogen bonds, responsible for inter–intra macromolecular association and thus film-forming ability. However, their hydrophilic nature entails poor moisture resistance and reduced capacity to hinder water vapour transmission [23]. To overcome these drawbacks, polysaccharides are modified through chemical pathways to obtain derivatives with enhanced performances or by blending them with hydrophobic materials and nanofillers.

##### Chitosan

Chitosan, or β-(l-4)-2-amino-2-deoxy-D-glucopyranose, is a cationic linear polysaccharide consisting of N-acetyl-glucosamine and N-glucosamine units. It derives from alkaline N-deacetylation of chitin, the second most abundant natural polysaccharide after cellulose. The primary sources of chitin are shellfish waste, insect cocoons, and fungi [44].

Chitosan is biodegradable, non-toxic, bio compatible, and broadly available. It is widely used for many applications in the biomedical, cosmetic, agricultural, and food sectors. The biodegradable property of chitosan results from the sensitivity of glycosidic bonds to chemical and physical breakdown, mainly due to oxidation and reactivity with enzymes (hydrolases), acids, and alkali compounds. Due to the absence of nearly positively charged amino groups, the A-A and A-D glycosidic sections are the preferred targets of hydrolysis in acidic conditions [45]. In general, it appears that as the acetylation levels increase, so does the degradation rate. This concept is true even for lysozyme, an enzyme present in human saliva and tears [46].

Chitosan is insoluble in water but soluble in acid aqueous solutions due to the protonation of the NH_2_ groups. It exhibits good antimicrobial activity against Gram-positive and Gram-negative bacteria, filamentous fungi, and yeasts [47]. 

Chitosan shows excellent film-forming abilities. However, extrusion technology is inadequate to produce chitosan-based films due to the low degradation temperature of this polymer and its non-thermoplastic behaviour. As a result, the production of films is mainly conducted through the solution-casting method. 

These films have good mechanical properties and effectively obstruct O_2_ and CO_2_ transmission [48]. Meanwhile, they are highly sensitive to moisture transmission, which compromises their use to preserve fresh or fatty food products. To overcome this criticism, authors investigated different strategies including chemical crosslinking and grafting with secondary components [49]. These methods provide an interpenetrated structural network to the resulting films, improving their hydrophobicity. Another suitable technique is blending chitosan with compatible polymers to induce a strong inter–intramolecular hydrogen bonding, which results in improved barrier and mechanical performances of the blend films [50].

##### Cellulose and Derivatives

Cellulose, or (1→4)-β-D-glucopyranosyl, is a linear chain polysaccharide in which anhydrous glucose rings ((C_6_H_10_O_5_) n) are bound through β1-4 glycosidic bonds, and the number of repeat units depends on the source material [51]. It constitutes the most abundant biopolymer in nature and can be degraded by cellulolytic microorganisms. In nature, the synergism between cellulolytic and non-cellulolytic microorganisms leads to the complete degradation of this polymer. These microorganisms are mainly aerobic and can synthesise cellulases enzymes (cellobiohydrolases and endoglucanases), which hydrolyse the β1-4 glycosidic bonds [52,53].

Native cellulose is water-insoluble due to its structural complexity, high crystallinity, and tightly packed hydrogen bonds, and is thus unable to form stable gels. This limitation is overcome by applying an alkali treatment followed by acidification using hydrophilic agents such as chloroacetic acid, methyl chloride, or propylene oxide to produce cellulose hydroplastic and thermoplastic derivatives. Cellulose derivatives are commonly isolated from wood, hemp, cotton, and other plant components [39]. These derivatives have been extensively investigated to develop biodegradable composites and films due to their high abundance, non-toxicity, and stability (Figure 2). 

Hydroplastic polymers obtained from cellulose are highly hydrophilic and possess excellent gelling capacity. They include carboxy methylcellulose (CMC), methylcellulose (MC), hydroxypropyl methylcellulose (HPMC), hydroxypropyl cellulose (HPC), and others [55]. Films and coatings based on these polymers are transparent, odourless, resistant to oxidation, and show enhanced mechanical and gas barrier properties [19]. However, they are highly sensitive to water vapour transmission due to their hydrophilic nature, which limits their application to dried and low-fat foods. In this context, several strategies have been investigated to confer hydrophobicity to cellulose-based films, thus reducing their WVP value. Shahbazi et al. [34] applied surface modification of CMC based films via reaction with sodium benzoate and glutaraldehyde vapour, followed by photo-crosslinking or chemical-crosslinking with gelatin. Authors observed that photo-crosslinking improved hydrophobicity and water barrier property more than the chemical crosslinking. Another study tested cellulose-based films obtained via chemical crosslinking of CMC with hydroxy ethylcellulose (HEC) using citric acid [56].

Cellulose acetate is the most researched thermoplastic polymer derived from native cellulose. This derivative is obtained treating technical-grade cellulose with a methylene chloride-acetic acid solution to substitute hydroxyl groups with acetyl groups [57]. FDA tagged cellulose acetate as GRAS, which prompted the food-packaging industry to develop and test novel applications of this polymer [54]. Cellulose acetate is commonly used to wrap fresh products and baked goods. Cellulose acetate films and coatings are tough and resistant to puncture. Conversely, they possess relatively poor moisture barrier properties, high rigidity, and lower thermal resistance compared with conventional thermoplastics [58]. These criticisms can be partially solved by adding plasticisers, which impart clarity and tailored rigidity. Moreover, when employed for prolonged applications, cellulose acetate may undergo partial hydrolysis to produce acetic acid [59].

##### Starch

Starch represents the primary energy reserve biosynthesised in the plants and one of the most plentiful renewable feedstocks. Native starch consists of two types of glucose polymers: amylose, a linear polysaccharide with (1→4)-α-D-glucopyranosyl units, and amylopectin, branched amylose with (1→6)-α-D-glucopyranosyl side units. Starch has been extensively studied as a biodegradable plastic and food hydrocolloid component thanks to its renewability, biodegradability, and excellent film-forming capacity. This polymer can be easily degraded in water, since amyloglucosidase or α- and β-amylase can form complexes with starch and hydrolyse the glycosidic linkages [60]. This process is strongly influenced by pH, the degree of crystallinity of starch, and its retrogradation [61].

Starch-based films and coatings exhibit remarkable mechanical strength, elasticity, transparency, and low oxygen permeability [15]. The major challenges related to native starch films are brittleness and high hydrophilicity, which results in poor water vapour barrier properties. These drawbacks preclude the application of starch-based films and coatings to package foods sensitive to moisture and oxidation [20]. To enhance the flexibility and water resistance, food-grade plasticisers (e.g., glycerol, glycol) and hydrophobic substances can be incorporated into the film-forming solution [47].

##### Pectin

Pectin is an anionic, hydro soluble, and high-molecular-weight heteropolysaccharide. It is one of the main components of the plant cell wall, contributing to tissue rigidity and integrity. 

Pectin is chemically composed by poly α-(1→4)-D-galacturonic acid chains [62], commonly known as homogalacturonan. Its linear structure is interrupted by rhamnose residues, on which secondary chains containing galactose, xylose, and arabinose are grafted. Consequently, pectin is composed of three different polysaccharide domains. The first domain is the homogalacturonan, which is the smooth component of the molecule. The second domain is named rhamnogalacturonan I and it is constituted by a chain of α-(1,2)-linked L-rhamnopyranose residues. The third one, rhamnogalacturonan II, is characterised by a complex and heterogeneous structure. The second and the third domains form the hairy regions of pectin [63] (Figure 3).

The carboxyl groups of galacturonic acid are partially esterified with methanol to form methoxylated groups, and can be converted to amide groups via reaction with ammonia [44]. According to the esterification degree (DE), pectin can be classified as low-methoxyl (<50%) and high-methoxyl (>50%) pectin. DE strongly influences the gelling properties of pectin [65].

The main industrial sources of pectin are orange pulp and apple pomace [47]. Pectin is widely applied in the food industry as a gelling, thickening, and stabilising agent for jam, drinks, and ice cream. It is recognised as safe (GRAS) by the FDA (2013) and it is well known for its biocompatibility, good gelling ability, and biodegradability. Degradation of pectin can be performed through physical (ultrasonication, radiation, photolysis, high-pressure treatment, etc.), chemical (pH differences of 3.5 allow either acid or alkali hydrolysis), and enzymatic processes (mainly pectate lyase, pectin lyase, and endo- and exo-polygalacturonase) [66,67]. 

The ability of pectin to form edible films and coatings has been largely investigated [63]. Some researchers suggested the scarce potential of pectin as a film-forming polymer due to its limited physicochemical and mechanical performances [68]. Despite that, several investigations have been conducted to improve pectin-based filming and coating properties. To enhance the mechanical stability of the film and the surface adhesion on the food substrate, pectin has been blended with food-grade plasticisers (e.g., glycerol, polyethylene glycol, and sucrose) and polymers (e.g., polyvinyl alcohol and cellulose derivatives). As well, pectin has been combined with hydrophobic compounds such as lipids to enhance its resistance to moisture and water vapour transmission.

#### 2.2.2. Proteins

Proteins are complex macromolecules characterised by variable molecular structures and exertion of different functional properties [69]. Protein derivatives are commonly isolated from natural resources and represent promising biopolymers to produce biodegradable packaging with excellent physicochemical, optical, mechanical, and barrier performances. In particular, the enhanced capacity of protein-based packaging to control gas transmission allows hindering the loss of flavours and restricting the migration of active components [70]. Besides, protein-based packaging can be easily degraded in the environment, and acts as a good biofertiliser due to the high nitrogen content [24].

The film-forming ability of protein derivatives strongly depend on their structure (e.g., sequence of amino acids, amount of intra-protein interactions), molecular weight, solubility, and charge [69]. Besides, proteins can be combined with other biopolymers, resulting in composite films with improved features [71]. 

##### Gelatin

Gelatin is a water-soluble protein obtained through the partial hydrolysis of native collagen, a primary component of bones and connective tissues of animals. This protein consists of a triple helix structure with repeated glycine-proline-hydroxyproline units. It is composed by a mixture of α-chains (one polymer/single chain), β-chains (two crosslinked α-chains), and γ-chains (three crosslinked α-chains), with relevant variability depending on the source [24]. According to the synthesis method, gelatin is broadly classified as (I) Type A, derived from acid-treated collagen, and (II) Type B, obtained from alkali-treated collagen.

Among biopolymers, gelatin has the peculiar capacity to form thermo-reversible gels with a melting point close to 40 °C. This attribute, along with the abundance, prompted its widespread use in food and pharmaceutical industries as stabilising agent and for the production of biodegradable packaging [29]. 

Gelatin-based films exhibit low O_2_ permeability and acceptable mechanical properties [72]. Additionally, gelatin can act as a carrier for natural antioxidants and antimicrobial agents. However, these films are highly sensitive to moisture and permeable to water vapour due to their hygroscopic behaviour. 

Numerous studies have been conducted evaluating the incorporation of crosslinkers, strengthening nanofillers, plasticisers, vegetable oils (e.g., corn, sun flower, essential oils), and natural polyphenolic antioxidants as promising methods to improve the performances of gelatin-based films and to support their bioactivity [42]. In particular, the cross-linking reaction was found to affect the intermolecular forces within the triple helix structure, resulting in an interpenetrated network structure of the film matrix (IPN) [27]. Moreover, gelatin has been blended with other biopolymers including chitosan [27] and zein protein [38] to produce a series of unique hybrid active films. Some studies have found that crosslinking reduces the biodegradability of gelatin. Instead, blending with highly hydrophilic polymers enhances the degree of degradability with respect to pure gelatin. In general, the molecular weight of gelatin typically affects the rate of degradation [27].

##### Corn Zein

Zein is a prolamin protein mainly isolated from corn seeds. It is an alcohol-soluble and biodegradable protein, whose hydrophobic nature relies on the high density of non-polar amino acids [73]. Moreover, it exerts a thermoplastic behaviour and outstanding film-forming properties [3]. These characteristics make zein a good candidate for the development of biodegradable packaging items. This protein can be easily degraded in specific environmental conditions (neutral pH, 50–60% of humidity, temperature over 40 °C) or in presence of proteases, such as trypsin, thermolysin, and pepsin [74].

Zein-based films are smooth, thermally stable, and possess low WVP values [75]. These attributes are mainly related to the formation of hydrogen and disulfide bonds between zein chains during solvent evaporation. For this reason, zein-based films can be tailored to act as selective barriers to oxygen, carbon dioxide, and oils. Despite that, these films generally exhibit poor mechanical properties and fragility, which can compromise their wide application. Thus, many strategies have been explored to improve their structural properties, including the addition of plasticisers and combination with other polymers to produce bilayer and composite films [15].

#### 2.2.3. Polylactic Acid (PLA)

Polylactic acid (PLA) is a compostable (under industrial conditions), biocompatible, and thermoplastic aliphatic polyester. This polymer can be completely degraded through a slow cleavage reaction of ester bonds. The process of biodegradation is carried out by microorganisms (*Actinomycetes*, other bacteria, fungi) or by degrading enzymes (proteases, cutinases, and esterases) [76]. 

PLA is obtained either through direct polycondensation of L- and/or D-lactic acid monomers or from the ring-opening polymerisation of lactide monomers. The first pathway is generally followed to produce low-molecular weight PLA, while the second method is applied to produce high-molecular weight PLA [20]. 

PLA is mainly synthetised by microbial fermentation from agricultural renewable sources such as corn, cassava, sugar beet pulp and sugarcane. Although 90% of total PLA is obtained by bacterial fermentation, the remaining 10% is synthetically produced by the hydrolysis of lactonitrile [77]. Currently, the annual production of PLA is estimated to be 140,000 tons, with an increasing trend due to its potential as a substitute for petroleum-based materials [78].

PLA properties include tensile strength, thermal stability, and gas permeability, and are comparable to those of synthetic polymers such as polypropylene, polyethylene, and polystyrene [30]. Moreover, PLA exhibits a better thermal processability compared with other thermoplastic biopolymers, and thus can be processed through conventional blow filming, injection moulding, fibre spinning, thermoforming, and cast filming [79].

PLA has been accepted as GRAS by the FDA [31]. As a result, this polymer has been increasingly employed in the food-packaging industry to produce disposable cutlery, plates, lids, and other items. Despite that, the high cost and the technical drawbacks, such as brittleness, low resistance to oxygen, and low degradation rate still deter the mass use of this polymer [3]. 

Considerable efforts have been made to improve PLA performances. Different blends of PLA with other natural biopolymers were tested. For example, blending with thermoplastic starch (TPS) enhanced the mechanical properties and the biodegradability rate of the biopolymer and reduced the production cost [37]. On the other hand, the PLA/PHB blend obtained by melt blending showed improved oxygen barrier and water resistance compared with pure PLA.

The addition of plasticisers represents another suitable strategy to improve the PLA mechanical performances. Thus, the demand for new “green” plasticisers based on natural and renewable resources such as vegetable oils is rapidly increasing [31].

#### 2.2.4. Polycaprolactone (PCL)

Polycaprolactone (PCL) is a semicrystalline biodegradable but non-renewable biopolymer of synthetic origin. This polymer is synthesised through the polymerisation of ε-caprolactone at high temperature (over 120 °C) or polycondensation of hydroxycarboxylic acid, yielding PCL with different degrees of molecular weight based on the alcohols used as catalysts. The final molecular weight affects the polymer’s properties: low molecular weight results in a crystalline, brittle, and hard film; high molecular weight results in a more elastic, tough, and poorly crystalline film [80].

PCL is characterised by its good solubility in organic solvents (i.e., chloroform, dichloromethane, benzene, tetrahydrofuran, toluene, etc.) at ambient temperature, insolubility in water, and partial solubility in other organic solvents, such as acetone, acetonitrile, ethyl acetate, and dimethyl formamide. However, the solubility in these last solvents can be enhanced through heat thanks to the low melting point temperature (60–65 °C) [81]. Although the physical and mechanical qualities are low and influenced by molecular weight, the barrier properties to oxygen and water vapour are excellent. These characteristics prompt the possibility to combine this polymer with others to improve its gas barrier properties for applications in food packaging. Therefore, PCL has attracted the attention of medical research due to its non-toxicity and potential applications in drug-delivery systems [82].

PCL is a biodegradable polymer that can be easily degraded through chemical and enzymatic hydrolysis thanks to the presence of ester groups [81]. The enzymatic method is preferable due to the rapid reactions that result in a complete polymer degradation in a few days [83]. The composting of this polymer is particularly efficient due to the heat of the process, which can support the biodegradation process, and to the enzymes (in particular, lipase, and esterases) generated by the microorganisms involved in the process [80].

#### 2.2.5. Polyhydroxy Butyrate (PHB)

Polyhydroxy butyrate (PHB) belongs to the family of the polyhydroxy-alkanoates (PHAs), a series of biodegradable, crystalline, and thermoplastic polyesters synthesised from microbial fermentation of organic biomass. It is produced by the Gram-positive bacterium *Bacillus megaterium* [25]. 

This polymer cannot be easily degraded by chemical treatments. Instead, it is more susceptible to thermo-mechanical degradation, oxidation, photodegradation, and enzyme and biotic degradation. The enzymes usually involved in this process are esterases, lipases, and proteases, which work through hydrolysis of ester linkage of the polymer. Biotic degradation is carried out mainly by PHB depolymerase, synthesised by *Alcaligenes*, *Pseudomonas*, *Comamonas* spp., and other species of bacteria, fungi, and algae [84].

PHB exhibits remarkable technical performances, comparable to those of polyethylene and polypropylene. Moreover, owing to its lamellar structure, it has superior water vapour barrier properties and a lower carbon footprint than conventional plastics. In fact, it is easily biodegraded by the action of PHA hydrolases depolymerases, which form (R)- and (S)-hydroxybutyrates and other non-toxic compounds under aerobic or anaerobic conditions [85]. 

These attributes make PHB a sustainable candidate for the replacement of fossil commodity polymers for short-term applications. Despite that, some criticisms, i.e., high brittleness, low thermal stability, and reduced processability still limit its widespread use [86]. Many attempts have been made to overcome these limitations. Arrieta et al. [87] blended PHB with PLA thanks to their comparable melting point temperatures, showing improved flexibility with respect to pure PHB. Additionally, extensibility can be enhanced by incorporating plasticiser or by fabricating composites through the addition of nanofillers [86]. 

## 3. Bio-Active Packaging

Food packaging has evolved beyond its use as simple containers and barriers against external factors. The consumer demand for healthy, safe, and more sustainable products has prompted scientists and industries to develop packaging materials able to actively ensure food safety and extend the shelf-life, thus maintaining food quality and taste [88]. This new packaging approach is known as “active packaging” [89]. 

Active packaging items are designed as “materials and articles that are intended to extend the shelf-life or to maintain or improve the condition of packaged food; they are designed to deliberately incorporate components that would release or absorb substances into or from the packaged food or the environment surrounding the food” (European regulation [EC] No. 450/2009).

Active food packaging expands the features of traditional packaging, including containment, protection, preservation, and communication, shifting from a passive defensive role towards an active role. It acts as a medium of interaction among product, environment, and packaging itself, altering the native environment of the packed product [90]. Depending on its functioning mode, active packaging can be classified under two major categories: scavenging and emitting systems. Scavengers are materials that absorb undesirable substances from the internal packaging environment, including moisture, oxygen, carbon dioxide, ethylene, and odours/flavours. Conversely, emitters are designed to discharge specific substances with desirable properties to produce a positive impact in the packaging headspace [91]. These active compounds can be either part of the packaging material or enclosed inside the package, separated from the packed food. The advantages related to the first solution are (I) no possible manipulation by the consumer, decreasing the chance of contamination; and (II) the packaging is produced with conventional equipment, decreasing the complexity of the process (Figure 4). Some substances commonly added to the packaging system are antioxidant and antimicrobial agents, enzymes, aromatic compounds, nutraceuticals, and pre- or pro-biotics. Among these, antimicrobial and antioxidant active compounds (either synthetic or natural-based) have been recognised as the most attractive ones to be incorporated into packaging systems, since microbial spoilage and lipid oxidation are considered as the two major causes of food deterioration [92].

### 3.1. Antimicrobial Packaging

Antimicrobial packaging has received increasing attention from food and packaging industries as a valuable alternative to thermal treatments to control the growth and avoid the spread of targeted pathogenic and spoilage microorganisms [20]. 

Concisely, antimicrobial packaging is obtained by incorporating an antimicrobial agent in the packaging material [18]. This represents a potential alternative to the direct addition of bioactive agents into or on the surface of food, which could lead to the immediate depletion of the antimicrobial functionality [94]. In this sense, antimicrobial packaging can exert a controlled release of the antimicrobial compounds, whose migration kinetics depend upon different factors such as the molecular structures of the polymer and antimicrobial compounds, the physicochemical characteristics of the packaging item, and the environmental conditions, both internal and external [18]. In this context, the design of an antimicrobial packaging system is complex, since it requires a thorough knowledge of five major factors: the food product; the internal package atmosphere; the targeted microorganisms; the packaging material; and the antimicrobial agent [95]. Different approaches have been explored for the development of bio-based antimicrobial packaging, as shown in Table 2 (Figure 5). According to their structure and production process, antimicrobial packaging systems can be categorised into five classes [91]. The first class consists of antimicrobial sachets which are included in the package, and gradually release the active compound during the storage period. In the second class, the active molecules are directly blended in the polymer matrix to produce antimicrobial items. The third class of antimicrobial packages are obtained by adsorbing a specific matrix, serving as a carrier of the antimicrobial additive, onto the packaging surface. This production method overcomes the disadvantages related to the second class, since the active compounds are not exposed to high temperatures and shearing forces related to the production process. In the fourth class, the antimicrobial agent is immobilised on the polymer matrix through ionic or covalent bonds between their functional groups. In this case, polymers and additives should share compatible functional groups, and the release of the active agent from the matrix largely depends on the type of bonding. The fifth class of antimicrobial packaging involves the application of polymers with intrinsic antimicrobial properties (e.g., chitosan). This approach requires direct contact between the packaging material and the food product for effective inhibition, which could be considered a limiting factor for two reasons: inhibition process is restricted to superficial contact layers; and the polymer must be approved as a food additive [96].

Antimicrobial compounds belong to several categories of molecules, either synthetic or extracted from plant, animal, and microbial biomasses [33]. All these classes of molecules have been successfully integrated into bio-based packaging, with promising results against pathogenic and spoilage bacteria and fungi [3]. 

#### 3.1.1. Essential Oils (EOs)

Essential oils (EOs) are aromatic secondary metabolites which are present in various plants. They consist in complex oily blends of 20–60 components, extracted from different plant parts including roots, leaves, flowers, and bark. They are extracted through solvent extraction, distillation, cold pression, and non-conventional technologies (e.g., microwaves; ultrasounds; supercritical fluids) [108]. The composition of EOs includes monoterpenes and sesquiterpenes as the predominant components, followed by phenolic acids, aldehydes, ketones, and terpenoids. Due to the presence of various active molecules, EOs have been reported to exert a broad number of biological activities [3]. 

The biocidal action of EOs is exerted through different pathways. However, it is commonly agreed that the main target of EOs is the cytoplasmic membrane of the microbial cell [109]. Since EOs are hydrophobic, their presence induces a change in the structure and fluidity of the cell membrane (Figure 6). This process triggers a cascade of chain reactions, resulting in internal pH disorder, electrical potential alteration, and impairment of the sodium-potassium pump, ultimately culminating in cell death [110]. 

Several studies, listed in Table 3, investigated the ability of EOs, either free or incorporated in biodegradable packaging, to impede the growth of Gram-positive bacteria (e.g., *S. aureus*; *L. monocytogenes*), Gram-negative bacteria (e.g., *Aeromonas hydrophila*; *E. coli*, *S. enterica*, *Campylobacter jejuni*, *Pseudomonas aeruginosa*) and fungi (*Fusarium* spp.; *Aspergillus* spp.; *Penicillium* spp.) [50,111,112]. These studies highlighted that the antimicrobial effectiveness of EOs depends on their specific composition and source, as well as the defensive strategies fielded by the microorganisms [18]. 

The most common EOs which are applied as active agents in food packaging include cinnamon (cinnamaldehyde) [123], rosemary [128], ginger [115], oregano [121], tea tree [35], citrus [122], and thyme [127] (Figure 7).

All these studies demonstrated that the presence of EOs can remarkably affect the structure of the packaging material, either improving or worsening the technical performances by interacting with the polymer matrix and the plasticisers [111]. Besides, their antimicrobial effect can be compromised by the fast release of volatile compounds. Furthermore, EOs may also influence the organoleptic attributes of foods [129]. A strategy to solve these issues is represented by the micro or nanoencapsulation of EOs and subsequent addition to the polymer matrix. This process allows performing a controlled delivery of the bioactive compounds and avoiding an excessive impact on the sensorial profile of food [121]. 

#### 3.1.2. Animal-Derived Polypeptides

Polypeptides are the most common animal-derived antimicrobial compounds. They are mainly secreted as a defence mechanism against bacterial spread [130]. 

Lysozyme is an animal-derived enzyme which was recognised as GRAS for direct inclusion in food matrices [131]. It is stable over broad ranges of temperature (4–95 °C) and pH (2–10). The biocidal activity of lysozyme has been tested against a wide range of pathogens and spoilage bacteria, finding its main effectiveness on Gram-positive bacteria such as *Clostridium tyrobutyricum* and *L. monocytogenes* [132]. 

Lysozyme expresses its antibacterial activity by disrupting the peptidoglycan layer of bacterial cell walls, achieved through the hydrolysis of the bond between N-acetyl-d-glucosamine and N-acetyl-muramic acid [133]. This specific mechanism makes lysozyme extremely effective against gram-positive bacteria, while the lipo-polysaccharidic layer of Gram-negative bacteria inhibits its access to the site of action. Many studies have suggested the possibility to expand the lysozyme activity by modifying its molecular structure through different pathways including covalent attachment of saturated fatty acids to lysine residues, thermal denaturation, glycosylation, reduction of disulfide linkages, and application of chelating molecules [134]. Nowadays, lysozyme is mainly used to challenge undesired butyric fermentation and late blowing caused by *C. tyrobutyricum* in semi-hard cheeses [135]. 

Lactoperoxidase is another animal-derived enzyme, secreted in the epithelial cells of the mammary gland and largely present in cow’s milk [136]. It is extremely effective against enteric bacteria including *Salmonella* spp., *Shigella* spp., and *E. coli.* It catalyses the oxidation of thiocyanate groups by hydrogen peroxide to yield thiocyanogen, which is then hydrolysed to hypothiocyanite. These unstable molecules react with the sulfhydryl groups of the bacterial cell membrane proteins, causing microbial death. This enzyme can be applied at ambient temperature, and thus is recommended for the preservation of raw milk [137].

Lactoferrin is a globular glycoprotein exerting antioxidant, anti-carcinogenic, anti-obesity, and antibiotic properties. It is found in secretions of humans and other mammalians and in colostrum milk [138]. The antimicrobial activity of lactoferrin is due to its ability to chelate iron, disrupting the external membrane of gram-negative bacteria. Along with the biocidal activity, lactoferrin exerts a bacteriostatic action, decreasing the microorganisms’ proximity to nutrients. It resulted as effective against many pathogenic bacteria such as *E. coli*, *Klebsiella* spp., and *L. monocytogenes* [139].

#### 3.1.3. Antagonistic Microorganisms and Bacteriocins

Some microorganisms and their metabolites can prevent the growth of others. This ability has attracted the attention of researchers and industries, eager to apply them as a “natural shield” to the growth of pathogenic and spoilage microorganisms in food. Nowadays, the application of “antagonistic microorganisms” and their derivatives for preserving food has become widespread, and it is commonly referred to as “bio-preservation” [140]. 

The prominent class of antagonistic microorganisms employed in food systems are the Lactic Acid Bacteria (LAB). LAB have been defined as GRAS by the FDA and have obtained the Qualified Presumption of Safety (QPS) by the European Food Safety Authority (EFSA) [141]. 

The use of LAB to compete against undesired microorganisms has been investigated, along with their ability to produce nutrients and metabolites with antimicrobial properties. Successful results were achieved by applying them to fruit and vegetables [142], fresh dairy products [143], and cooked meat [144]. In these studies, different species of *Lactobacillus* showed their capacity to thrive in competition with bacterial (e.g., *L. monocytogenes*) and fungal (e.g., *Penicillium* spp.) populations.

Bacteriocins are proteinaceous metabolites mainly produced by LAB as a defence mechanism against other microbial strains. Their promising application has been assessed on a wide range of food products, including minimally processed fruits and vegetables, dairy products, meat and fish. In particular, their maximal potency is expressed when combined with other technologies through a hurdle approach [110]. Nisin and pediocin are the major bacteriocins that have received attention as promising food bio-preservatives [133].

Nisin is a heat-stable protein produced by specific *Lactococcus lactis* strains. It possesses a strong antibacterial activity against Gram-positive bacteria such as *Staphylococcus*, *Bacillus cereus*, *Clostridium* spp., *L. monocytogenes*, and others. However, it exhibits a lower inhibiting activity against Gram-negative bacteria and fungi [145]. In fact, nisin hinders the growth of Gram-positive cells by binding to specific groups of the cell wall, which results in the poration of the cell membrane and the loss of intracellular constituents [146]. Nisin found one of its most promising applications in controlling the populations of *L. monocytogenes* and *Clostridium* spp. in dairy products [147]. 

Pediocin is produced by different species of *Pediococcus*, a group of Gram-positive, homofermentative bacteria belonging to the family of *Lactobacillaceae*. Pediocin acts by generating holes in the cytoplasmic membrane of the target cells, reducing the intrinsic pH and inhibiting the proteins responsible for energy production [148]. The addition of concentrated pediocin has been tested for the preservation of vegetables, dairy products [149], and processed meat [150]. The activity of pediocin in food is mainly influenced by pH, osmotic equilibrium, enzyme activity, and temperature. 

Bacteriocins have been applied as antimicrobial additives incorporated in active packaging. For example, nisin has been successfully employed in antimicrobial films (both petroleum-derived and bio-based), used to wrap raw and processed meat, and tested against *Listeria* spp. [97]. Moreover, its impact on the technical properties of biodegradable films was evaluated in a recent study [103].

### 3.2. Antioxidant Packaging

Antioxidant packaging represents another trend category of active packaging. In this case, packaging is enriched with active compounds able to delay the oxidation rate of the packed products [16]. 

With respect to the food sector, the activity of an antioxidant agent is mainly addressed to suppress the ignition of lipid oxidation chain reactions, which naturally occurs within biological matrices. This process causes the gradual alteration and decay of colour (enzymatic oxidation), odour, and flavour (oxidative rancidity), structure (softening), and nutrients [151]. Antioxidants strongly differ from each other for their reaction pathways. Some molecules act as “direct” antioxidants, reacting with intermediate peroxyl radicals and blocking the subsequent reactions (e.g., glutathione, ascorbic acid, polyphenols). Other molecules act as “preventative” antioxidants, binding cationic metals such as Fe (II) and Cu (II) (e.g., albumin) [4]. According to their molecular nature and reactive mechanism, antioxidants can be employed to produce release-type packaging, which transfers the active substance to the food surface at a sustainable rate, or scavenging-type packaging, which sequesters target radicals and ions without affecting the food composition [23]. 

The development of an antioxidant packaging system starts with the selection of the bioactive agent, which must comply with two requirements: (I) suitability for the target product to-be-preserved, and (II) compatibility with the polymer matrix to achieve a homogeneous distribution of the substance in the packaging item [152]. Focusing on bio-based and edible packaging, antioxidant films and coatings are mainly obtained through direct incorporation of the active molecule in the biopolymer matrix. Other techniques involve the functionalisation of the packaging material via physical (e.g., encapsulation) or chemical (e.g., crosslinking, plasticiser addition) processes, which affect the adhesion of the active compounds to the polymer matrix. These processes allow tailorising the rate of release and/or the scavenging mechanism of the active molecule, adapting the materials for a broad range of applications [153].

A broad variety of antioxidants have been evaluated for the development of active packaging, as shown in Table 4. The current trend is focused on replacing synthetic additives with natural and harmless alternatives. 

#### 3.2.1. Natural Antioxidants

Natural antioxidant molecules can be mainly categorised into three sub-groups: (I) vitamins (e.g., ascorbic acid; α-tocopherol), (II) carotenoids (e.g., carotenes; xantophylls), and (III) phenolic compounds. 

Polyphenols constitute the most popular and important group of naturally occurring antioxidant compounds employed for the production of active packaging due to their strong free-radicals scavenging effect [154]. 

The antioxidant activity of polyphenols is commonly ascribed to single-electron transfer and hydrogen transfer mechanisms, which allow the active molecule to react with active radical species of the matrix, producing stable and harmless oxidised molecules. 

Related to their composition, polyphenols can be classified into (ii) non-flavonoids and (ii) flavonoids. Among them, flavonoids are the most largely studied for packaging applications due to their strong antioxidant activity. Flavonoids are present in the form of flavonols, flavones, isoflavones, anthocyanins, and others. Most of them are polar, which makes them compatible with most of the hydroplastic polymers, and extracted through protic solvents (e.g., water, ethanol, methanol, isopropanol) from non-edible portions of fruit and vegetable by-products, such as peels and seeds [155].

Generally, polyphenols are not employed in active packaging singularly, but mostly exist as complex mixtures which include aqueous and alcoholic plant extracts, essential oils from spices and herbs, and a broad variety of phenolic concentrates obtained from various waste bio-sources [156]. For this reason, the overall antioxidant activity of these products not only refers to their polyphenolic content, but it strongly depends on their source, chemical composition, and extraction process [157].

#### 3.2.2. Plant Extracts

The inclusion of plant extracts, as complex systems containing numerous molecular components, has the potential to functionalise bio-based packaging materials with antioxidant bioactivity. These mixtures are isolated from several botanical sources through solvent-extraction technology. The extraction efficiency, and thus the phenolic content of the extracts, can be varied by changing the operational parameters, such as time, temperature, solvent type, solvent concentration, and pH [155]. Moreover, physical processes such as microwave, ultra-sonication, and milling allow further enhancing the extraction rate of these antioxidants [158]. 

The main vegetal sources of polyphenolic extracts used in food packaging comprise medical plants (leaves, roots, and stems), and various parts of fruits and vegetables. Among medical plants, extracts from thyme [36], black tea [159], green tea [160], mint [161], rosemary [114], and sage [162] have been added to film-forming solutions to produce antioxidant films for packaging purposes. Edible fruits, grape seed [104], pomegranate peel [163], thinned apple [164], and others have been evaluated as sources of polyphenolic extracts. All these studies highlighted the ability of the extracts to enhance the radical scavenging capacity of the polymer, mainly due to their high phenolic component. 

The polyphenolic prolife of an extract strongly changes in relation to its source. According to their composition, different extracts diversely interact with the polymer matrix, creating variable hydrogen-bonding patterns [160]. This fact not only influences the final antioxidant property of the film but can alternatively affect the mechanical and barrier properties of the packaging item. For example, in some cases the large number of viable hydroxyl groups induce an increase of the free volumes in the blend matrix, leading to highly flexible films [114]. In contrast, the rigid aromatic and heterocyclic rings of flavonoids can act as physical crosslinkers of the polymer chains, improving the tensile strength and elastic modulus of the film [2].

## 4. Nanotechnology in Biodegradable Packaging

Nano-technology represents one of the major research topics of the packaging sector due to the huge number of prospective applications and advantages [165]. 

The use of nano-materials traditionally covers many aspects of the food sector, including food safety, nano-sensors, nutrients delivery, and pathogen detection [4]. Lately, nano-technologies have been utilised to improve the technical performances of conventional bio-based materials, and to give them additional features. Besides, this novel approach is laying the basis for the development of a new generation of smart and intelligent food packaging systems, able to localise, sense, and remote control the food items [166]. 

The use of nano-structures (i.e., nano-fillers, bio-nanocomposites, and nano-capsules) is expected to broadly enhance the potentialities of bio-based packaging, and extend the number of smart packaging solutions in the next few years. 

### 4.1. Bio-Nanocomposite Materials

Nanoparticles are characterised by nanoscale dimensions, usually <100 nm. When nanoparticles are incorporated into a biopolymer material with specific technological purposes, they take the name of “nanofillers”, and the resulting item is called a “bio-nanocomposite” [167]. Bio-nanocomposite materials may be defined as a multiphase material in which a continuous phase (i.e., a biopolymer) is embedded with a non-continuous nano-dimensional phase (i.e., a nanofiller), either inorganic or organic [168]. 

Due to their small size, high aspect ratio, and large interfacial areas, nanofillers have been firstly explored as structural reinforcing agents, with the function to improve the technological properties of packaging materials. When uniformly distributed in the polymer matrix, nanofillers are able to interact with the polymer chains, creating a tangled network of hydrogen bonds that fill the free spaces within the matrix and restricts its molecular mobility [91]. In this way, nanoparticles provide an overall enhancement of the mechanical, barrier, and thermal properties of the material with respect to traditional non-composite systems [169]. In particular, it was demonstrated that low concentrations of fillers (<5%) are able to significantly improve biopolymer properties, which is economically advantageous in view of their large-scale application [170].

Along with the structural function, the incorporation of nanofillers also represents a suitable strategy to confer additional functions to the packaging material. On the one hand, nanofillers can serve as bioactive additives, since some of them exhibit inner antimicrobial, antioxidant, and scavenging properties [166]. On the other hand, nanofiller incorporation can tailorise the retainment and release kinetics of bioactive compounds from the polymer matrix, and adapt the barrier performances of the packaging item [152]. As a result, the correct selection of a nanofiller (nature, quantity) and suitable process parameters to customising the bio-nanocomposite materials for countless potential applications [171]. Some case studies are shown in Table 5.

#### 4.1.1. Nano-Clays

Clays have gained remarkable interest as reinforcing fillers to improve the mechanical, thermal, and barrier properties of biopolymers [173]. These siliceous compounds mainly exist in the form of laminated one-dimensional (1D) or two-dimensional (2D) fibrous structures that can be easily dispersed into a polymer through two possible mechanisms, namely intercalation or exfoliation [175]. The latter mode represents the best strategy to incorporate these compounds into a polymer matrix, since it allows the complete delamination of the particles and their homogeneous diffusion [165]. 

Some widespread nano-clays applied to develop bio-nanocomposite materials are montmorillonite, bentonite, palygorskite, and sepiolite. Among these, montmorillonite have been largely tested due to its excellent technical behaviour, abundance, low cost, and compatibility with a wide range of biopolymers [176]. It consists of a hydrated aluminium silicate layered structure, with a modest negative charge which varies from layer to layer [175]. It possesses a high surface ratio and interfacial area, which contributes to its uniform distribution. 

The features of a clay-reinforced film strongly depend on the polymer matrix, nature of the clay, the clay–polymer interactions, and the processing conditions [102]. Besides, surface-modification methods have been tested on clays to enhance their capacity of interfacial interaction, including the use of alkylammonium cation surfactants. However, these surfactants are not appropriate for modifying clay surfaces in bio-applications due to their toxicity [177]. As a result, most clay-composites are prepared using unmodified clay materials. 

#### 4.1.2. Metal Nanoparticles

Metal nanoparticles such as copper (Cu), silver (Ag), gold (Au), and their alloys have been widely applied to produce nanocomposite active films and coatings due to their strong antimicrobial activity [178]. 

Different mechanisms have been postulated to explain the antimicrobial action of metal nanoparticles. In particular, Tamayo et al. [179] suggested a three-step mechanism to explain the antimicrobial activity of Cu-nanoparticles on the bacterial cell in Cu/polymer nanocomposites: (I) the biopolymer gradually releases Cu^2+^ ions, which permeate the cell wall and interact with the membrane proteins and lipopolysaccharides; (II) the cell wall collapses due to the weakening of the membrane, which leads to the loss of cell organelles; (III) ions interact with the bacterial DNA, causing its rupture and producing reactive oxygen species (ROS), which lead to oxidative damage and bacterial death (Figure 8). A similar mechanism was also proposed to describe the activity of Ag-doped edible packaging [178].

Despite their antimicrobial activity, metal nanoparticles possess a certain antioxidant activity, exerted via the radical scavenging mechanism [180]. Moreover, they are compatible with various natural antioxidant extracts and EOs, and thus can be used in synergy with them to produce films with enhanced performances [181]. Additionally, the incorporation of metal nanoparticles can alter the barrier properties of the material by filling the voids in the porous matrix [99].

#### 4.1.3. Metal Oxides

Metal oxides have been extensively studied for food-packaging applications due to their strong antimicrobial properties, which makes them promising alternative to organic agents. They include titania (TiO_2_), silica (SiO_2_), magnesium oxide (MgO), zinc oxide (ZnO), and others. Among these, TiO_2_ and ZnO are the most widely tested in the food packaging sector due to their specific physicochemical characteristics, chemical stability, and biocompatibility [182]. These nanoparticles have been tested both as a reinforcing agent to improve the technical properties of edible films, and as antimicrobial additives. Specifically, they possess a remarkable photocatalytic activity in the near-UV region, since they generate reactive oxygen species (ROS) that can directly damage the cell walls [183]. 

As an example, Siripatrawan et al. [101] developed TiO_2_-enriched chitosan films. The authors showed that increasing concentrations of TiO_2_ enhanced the photodegradation rate of ethylene. Besides, the film exhibited broad antimicrobial activity against Gram-negative and Gram-positive bacteria.

#### 4.1.4. Bio-Nanofillers

Bio-nanofillers are ultrathin structures produced by different methods (e.g., electrospinning; acid hydrolysis etc.) [184,185] from organic materials. They are biodegradable, renewable, and possess a high surface-to-volume ratio and low density. These particles have been extensively tested in the food packaging sector as reinforcing agents, and to modulate the delivery of bioactive compounds [186]. 

Cellulose derivatives are the most widespread bio-nanofillers to fabricate biodegradable composites [187]. Cellulose nanoparticles can be classified into three types, related to their structure: (I) cellulose nanocrystals (CNCs), which are rod-like crystals with 5–70 nm width and 100–250 nm length; (II) cellulose nanofibre (CNFs), which possess a fibrous structure with a width of 5–60 nm and length of several nanometers; and (III) bacterial cellulose (BNCs), which consists of ribbon-shaped fibrils with 70–80 nm width [188]. 

Many researchers have focused their attention on the extraction of nanocellulose from different sources of biomass and wastes, such as agricultural wastes, forest residues, algae residues, and industrial by-products [189,190,191]. The extraction methods can be divided into three different kinds of treatments: chemical, physical, and biological [189].

The chemical method represents the most conventional way to extract nanocellulose. It employs a bleaching treatment (e.g., oxidation by NaClO in water at pH10, in presence of NaBr and TEMPO for catalysts), alkaline treatment (80 °C for 2 h, 4.5% *w/v* NaOH), and acid hydrolysis (45 °C for 40 min, H_2_SO_4_ 60–64% *w/v*) [192,193,194]. The physical method represents an effective treatment, which allows obtaining the highest yields of extraction. The main drawback related to this method is that it is highly energy-consuming. Grinding, homogenisation, ultrasound, high-pressure, and screw extrusion processes are widely employed for this purpose [189]. Last but not least, the biological approach involves the treatment of the cellulosic matrix through microorganisms, which can synthesise enzymes for the degradation of cellulosic materials [189], or through the direct application of cellulases enzymes (such as cellobiohydroalases and endoglucanases) [185,190].

The combination of these techniques can overcome the drawbacks related to every single method. 

BCN is produced mainly by *Komagataeibacter xylinum* (but also *Agrobacterium tumefaciens*, *Dickeyadadantii*, *Salmonella enterica*, *Pseudomonas putida*, *Rhizobium leguminosarum*, *Escherichia coli*) bacteria, through molecular pathways that involve the presence of glucose or different other sources of carbon [195]. 

Cellulosic nanofillers exhibit a characteristic self-association property, deriving from the inter- and intramolecular hydrogen bonding involving their surface hydroxyl groups [196]. This promotes the strong adhesion of these materials on and within the polymer matrix, enhancing the mechanical characteristics of the composite material by creating tortuosity, crystal nucleation, and chain immobilisation [105]. In addition, the highly tortuous structure induced by crystalline fibres can hamper the water vapour diffusion, resulting in low WVP values. Due to their surface reactivity, they can also serve as bio-scaffolds.

Cellulosic nanoparticles possess an enormous amount of active surface hydroxyl groups that can be modified by chemical reactions such as cationisation, silylation, carboxylation, polymer grafting, and hybridation with metals and metal oxides [23]. In particular, surface-modified nanofillers possess higher interfacial compatibility with a range of biopolymers compared with un-modified ones (Figure 9). Surface-modification also influences the polarity and hydrophilic behaviour of the material, enhancing its ability to hinder vapour diffusion throughout the packaging system [197].

Oyeoka et al. [198] demonstrated the fast water absorption rate and biodegradation of films incorporated with cellulose nanocrystals (CNCs). The behaviour of the CNCs at different concentrations was interesting: at lower levels of incorporation, the films tended to absorb more water (until 516% in 50 min) and be more resistant to degradation in soil; conversely, at higher levels of CNCs, the degree of absorption of water decreased (until 373% in 50 min), and the resistance to degradation was slightly reduced.

### 4.2. Nano-Encapsulation and Nano-Emulsions

Encapsulation is a technology which consists of packing a target substance into a solid envelope, with the double purpose to protect it from external interactions and provide a controlled release under specific conditions [125]. Commonly, hydrophilic materials (i.e., polysaccharides, proteins) are used to encapsulate hydrophobic substances, and vice versa. 

According to their size, capsules can be categorised as macro-, micro-, and nano- [117]. Specifically, nano-capsules have been widely applied as carriers of nutraceuticals (macronutrients, enzymes, prebiotic, probiotic, vitamins, omega-3-fatty acids) and technological additives (antioxidant, antibacterial, and antifungal chemicals; colourants; flavours) to produce functional food with enhanced safety and stability [199]. In addition, they were used to dope biodegradable films and coatings to fabricate nanocomposite packaging solutions [118]. As an example, Liu et al. [200] developed films based on gelatin and enriched with different concentrations of tea polyphenols/chitosan nanoparticles. The incorporation of nanoparticles decreased the WVP of the resultant films. Moreover, the release kinetics of tea polyphenols from the film surface were evaluated by means of two food simulants (i.e., 50% ethanol at 4 °C; 95% ethanol at 25 °C). The study highlighted a slow releasing rate of the polyphenols for both the simulants, which was probably due to the film’s tortuosity and increased diffusion pathways induced by the nanoparticles. Similar results were obtained by Cui et al. [201] for zein films doped with pomegranate polyphenols/chitosan nanoparticles.

Nano-emulsification represents another technique which allows to increase the bioavailability and stability of bioactive compounds, and to guarantee their proper delivery in the surrounding environment [202].

A nano-emulsion is a system composed by two immiscible liquids in which one is homogenously dispersed in the other, forming nano-sized globules (50–500 nm). Due to their high ratio of droplet surface/mass unit, nano-emulsions possess a high delivery/encapsulation ability [203]. 

The most widespread application of nano-emulsions in the food industry consists of the retainment and controlled delivery of active agents to solid foods. Bioactive molecules such as EOs can be directly incorporated into a food system or entrapped in polymer matrices to produce active packaging [50]. In particular, the incorporation of nano-emulsified EOs into biodegradable materials has the double advantage to minimise the concentration of active agent required to perform a valuable antimicrobial activity and to reduce its sensory impact (Figure 10). 

## 5. Biodegradable Packaging from Agri-Food Waste

To date, about 30–50% of food is wasted from post-harvesting to processing, storage, and consumer usage. Typical examples of food by-products are vegetable peels, fruit pomace, seeds, and low-quality whole fruits and vegetables [204]. The large part of these matrices is still discarded in landfills, while a small portion is valorised for bioprocessing [205].

A feasible strategy to valorise food waste and by-products consists of their use for the production of bio-based packaging materials. This approach involves two remarkable benefits. On the one side, food by-products constitute a cheap, renewable, and under-utilised source of polysaccharides, lipids, proteins, and many other components [206]. These components can be either employed as the major constituent of packaging or as minor additives, resulting in the reduction of the production costs. In particular, the inclusion of by-products components has been demonstrated to improve the engineering properties of the packaging material, thus conferring it additional activity [207].

### 5.1. Life Cycle Assessment LCA

According to estimates by “Plastic Europe 2022”, the packaging sector is responsible for 33.5% of plastic consumption worldwide, but only 6.6% of this plastic is recycled [208]. Due to this circumstance, it is necessary to determine the carbon footprint of these materials to obtain an in-depth report about possible environmental damage to both production and potential recycling.

For this reason, is important not only to replace non-biodegradable materials, but also to estimate through a holistic approach the impact on the environment of the new compounds chosen to be the new green polymers [209]. LCA is an analytical and systematic methodology to estimate the ecological footprint of the entire process of modification, transformation, transportation, emissions, and waste of a product. Life Cycle Assessment, regulated by ISO 14040:2006, has developed into a legitimate area of study in the field of research, becoming mandatory for efficient organisational, commercial, and disposal process analysis.

Numerous studies have been recently conducted to highlight the issues related to the production of plastics from non-renewable and poorly biodegradable sources [210,211,212]. These issues should be understood and exploited with a view towards a greener recycling process for petroleum-based materials, with the hope of eventually replacing them entirely.

### 5.2. Pre-Treatments of By-Products and Application for Packaging Production

The most common way to prepare bio-based packaging containing food by-products involves to directly blend the whole by-product or its components with biopolymers and additional additives [213]. A necessary step to apply by-products for packaging production is represented by their pre-treatment. 

The first step of pre-treatment usually involves drying and milling processes. The drying stops the microbiological decay and enhances the handling of the product. Milling process reduces the size of the product particles, improving the processability, uniformity, and dispersibility for blending [214]. 

A further step of treatment can involve the isolation of specific components through conventional or non-conventional (e.g., microwave or ultrasound) extraction techniques [75]. This process allows isolating and purifying specific fractions of the raw material, which is subsequently added to the film-forming solution for specific purposes such as technical properties enhancement (e.g., polysaccharides to improve mechanical properties; lipids to improve water-barrier properties) and providing additional features to the packaging material (e.g., polyphenols for antioxidant capacity; essential oils for antimicrobial activity). 

In recent years, some novel approaches have shown their potential as valuable ways to valorise food by-products for packaging development. Among these strategies, it is worthy to cite the extraction of fibres and cellulose from different by-products [215], the isolation of nano-sized cellulose and their employment to improve the mechanical and water-related properties of packaging material [174], the production of cellulose by bacteria from different foods by-products [216], chemical modifications of the raw material by different methods (e.g., grafting) [217], and fermentation of fruit juice pulp to obtain thermoplastic biodegradable polymers such as poly-hydroxy alkanoates [218] (Figure 11). 

### 5.3. Impact on the Engineering Properties of Packaging 

In recent years, particular attention has been dedicated to by-products (both whole and fractionated) as sustainable and green bio-fillers to produce materials with enhanced technical characteristics. 

Taking into account the mechanical properties, Nair et al. [32] showed that inclusion of 5–15% of wood-based CNCs led to a significant increase in the tensile strength of PLA films, mainly ascribed to the densified volume fraction of fibrils. Yang et al. [174] observed that, according to the treatments performed on nano-sized cellulose (e.g., presence/absence of solid-state shear milling), their addition to the polymer matrix can either decrease or increase the tensile strength of the final film. This effect mainly depends on the interfacial contact area achieved between the nano-sized fibres and the polymer chains. Overall, it is interesting to note that many bio-based materials enriched with nano-sized cellulose have tensile strength comparable to commonly used low-density polyethylene (7.0–25.0 MPa), while the elongation percentage of most films are significantly lower. Besides cellulose-based bio-fillers, other compounds derived from by-products can help improve the mechanical properties of packaging materials. As an example, pomegranate peel extract was found to enhance the elongation percentage of protein-based (from 81% to 173%) and PVA films (from 48% to 182%) based on the polyphenol interaction with the material matrices, which chemically strengthened the composite [213]. 

The addition of by-products can reduce the water vapour permeability of a packaging material by altering its overall hydrophilicity (reducing the available hydrogen groups) and the structure of the biopolymer (increasing the tortuosity for the passage of water molecules). 

Grape seed extract [219], lime peel extract [207], and other extracts were found to improve the water barrier properties of the tested materials when applied at specific concentrations (excessive or not sufficient concentrations can either have no significant effect or worsen the properties). 

Aside from the above-mentioned properties, some researchers highlighted the changes in oxygen barriers, optical properties, thermal properties, and the morphology of bio-based materials induced by the addition of food by-products. For example, the introduction of discarded balsamic vinegar or tea leaf waste extract remarkably decreased the oxygen permeability of PVA films [220]. As well, thermal stability could be enhanced by strengthening the chemical bonding pathways within the biopolymer matrix [219], or by including high thermal-stable components such as lignin [32]. 

### 5.4. Impact on Antioxidant and Antimicrobial Capacities of Packaging

Food by-products contain a large number of bioactive compounds (i.e., polyphenols, organic acids, EOs). Recently, the application of these compounds has caught the interest of many researchers as an appealing strategy to confer targeted capacities to packaging systems. 

Regarding the antioxidant activity, the addition of pomegranate peel extracts [213] apple skin powder [98], and black plum peel extract [75] resulted in a significant increase of the antioxidant capacity of the final films. Some researchers applied various by-product extracts to prevent the oxidation of lipid-rich foods. As an example, chitosan films enriched with olive pomace resulted in significantly lower peroxide values in walnuts compared with control (without extract) and polyethylene plastic films after 31 days of storage [221]. 

Along with antioxidant capacity, various by-products can also confer antimicrobial properties to the packaging material, especially in the form of extracts. Two examples are pine needle extract [106] and black plum peel extract [75]. 

The variable antimicrobial activities of extracts from by-products mainly result from the mixed active compounds that characterise their specific composition. Moreover, the antimicrobial efficacy strongly depends on the applied concentration and the interaction with other components, which can be either synergistic or contrasting. 

## 6. Future Challenges and Concluding Remarks

Much effort has been devoted to developing bio-based active packaging solutions (Table 6). However, there is still a deep gap between laboratory-scale research and real-time applications and commercialisation. 

The first root of this gap is technological. It is worthy to cite some practical examples:EOs possess a strong biocidal efficacy on a broad range of microorganisms, which virtually makes them suitable alternatives to conventional preservatives. However, each of them also possesses a peculiar aromatic profile, which could negatively affect the flavour of food, and this hinders their broad usage [222];Most of the biodegradable packaging films still do not provide a sufficient water barrier for moisture-sensitive foods, and so their feasible applications are mainly restricted to disposable food wrappers for fast foods that do not require improved water barrier properties [174];To date, most of the studies focused on packaging with antimicrobial and antioxidant properties are still performed at the in vitro level. For the future, it would be worthy to extend the achieved findings to in vivo experiments in order to provide the food industry with more specific data about the impact of these extracts on food safety, quality, and shelf life.Industrial production of biopolymers for the replacement of plastic is still an impossible path to pursue due to the cost of production of these molecules compared with plastics.

The second root is economical. For example, authors suggested reducing the sensory impact of EOs by entrapping them in nano-emulsions. However, these approaches suffer economic restrictions since they are not cost-effective [223]. Besides, talking about natural metabolites, their production relies on the availability of their resource, on the extraction procedure, and the purification steps, etc. All these aspects contribute to increase the final price.

The third root is related to the impact of these compounds on human health and the environment. In this sense, the composition of each active agent, its specific migration rate from the packaging material, and the interactions with the food product should be fully characterised to avoid any possible hazard for human health and to ensure the quality of the whole package. This is particularly the case of nano-technology application in food packaging [23]. The risks related to nanomaterials are mainly due to the lack of knowledge about their mechanisms of migration from the packaging to the food product and the environment. In this sense, food regulatory bodies such as FDA and EFSA have expressed their reservations about the extensive application of these materials and established strict regulations on the transfer threshold of these compounds. For example, EFSA established that the upper limit for silver migration in food packaging is 0.05 mg/L in water and 0.05 mg/kg in food (EFSA, 2021). In this sense, further and in-depth research about the migration pathways of these particles is strictly required to sustain their regulatory approval [167]. 

These concerns are certainly a significant drawback for the pilot and industrial exploitation of natural compounds as additives in novel, upgraded, bioactive food packaging materials. However, environmental pollution connected to the disposal of foods, agro-industrial by-products, and conventional plastic packaging are becoming significant issues. For this reason, it is necessary to encourage research in the field of biopolymers based on sustainable production (i.e., use of by-products from the industry as extracting matrices; utilisation of green solvents and physical treatments; microbiological processes) to fulfil the market demands and to achieve the goals outlined in the 2030 Agenda (UN).

Overall, further efforts will be needed to strengthen our knowledge about all the branches of this field. These novel studies will allow the green era of food packaging to move a step forward towards the future. 

## Figures and Tables

**Figure 1 polymers-14-04257-f001:**
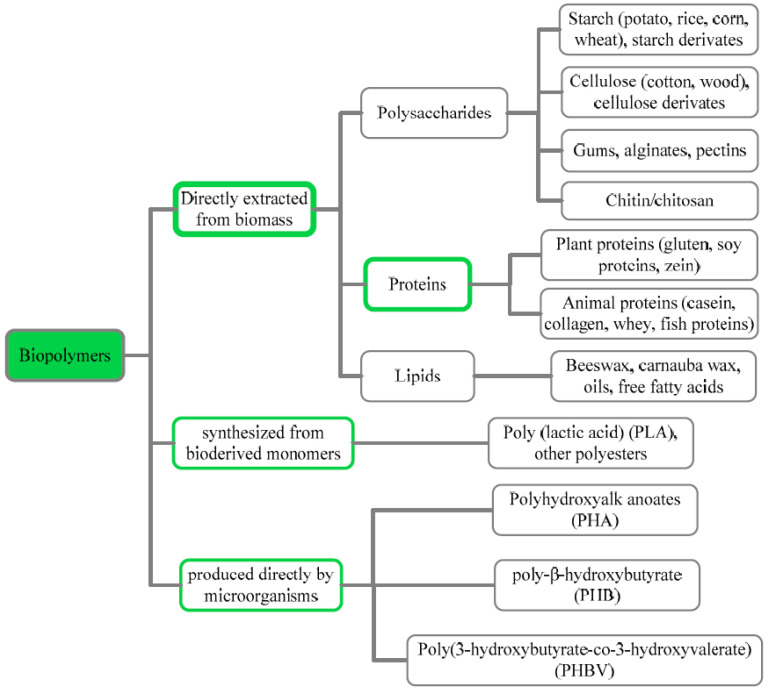
Classification of biopolymers (reproduced with copyright permission from Chen et al. [24]).

**Figure 2 polymers-14-04257-f002:**
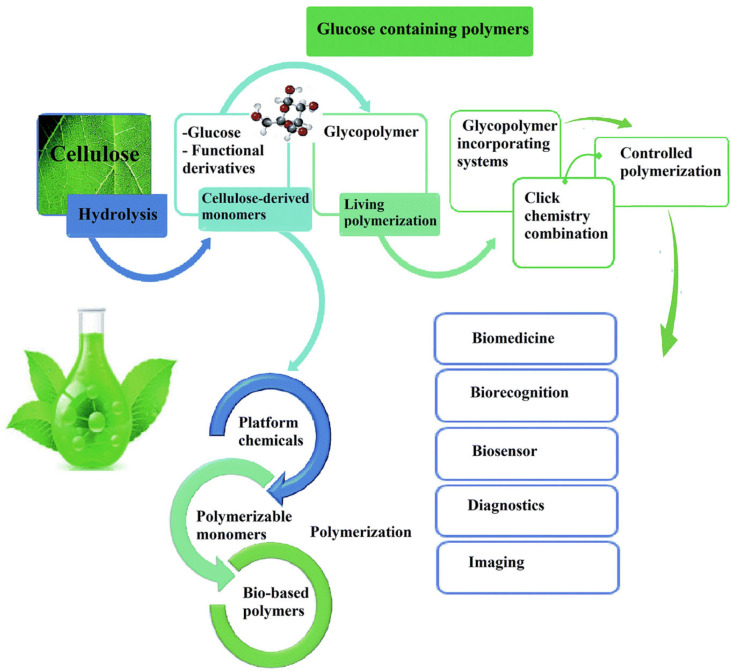
Examples of possible applications of monomers of cellulose for polymer production (reproduced with copyright permission from Shaghaleh et al. [54]).

**Figure 3 polymers-14-04257-f003:**
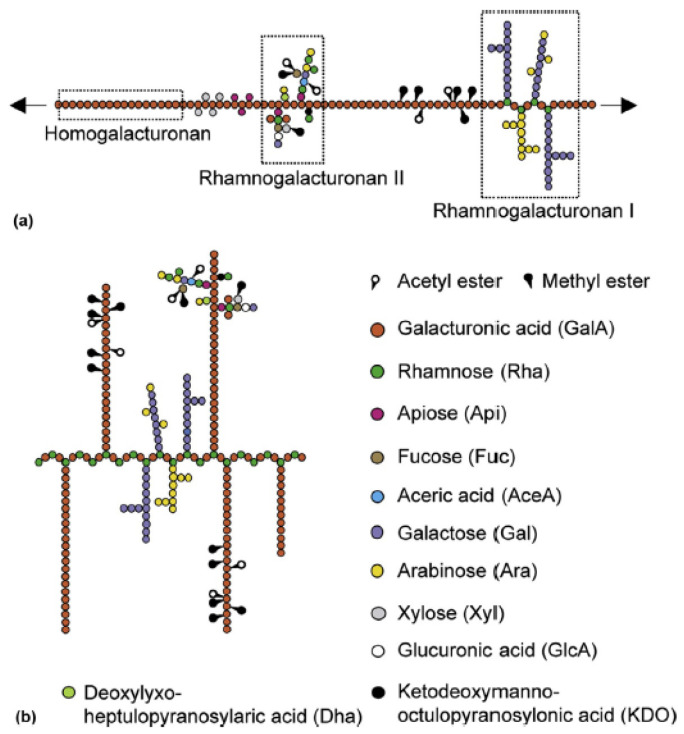
Comparison between (**a**) the traditional and (**b**) the modern pectin model (reproduced with copyright permission from Willats et al. [64]).

**Figure 4 polymers-14-04257-f004:**
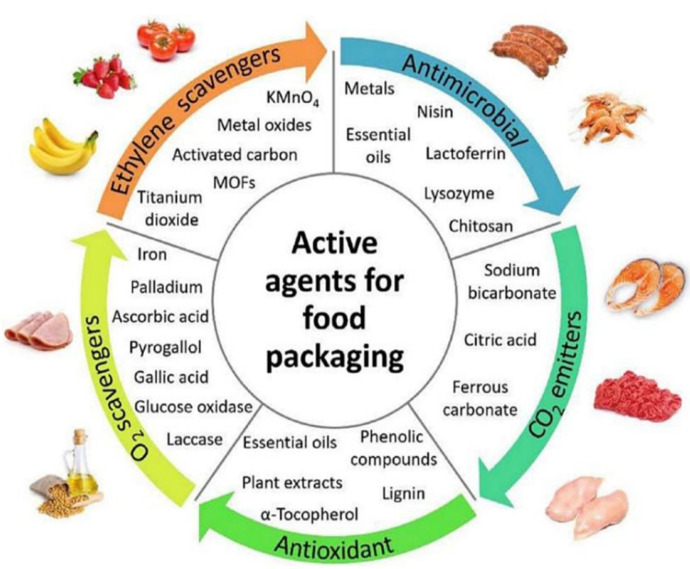
Application of different bioactive compounds in active packaging (reproduced with copyright permission from Vilela et al. [93]).

**Figure 5 polymers-14-04257-f005:**
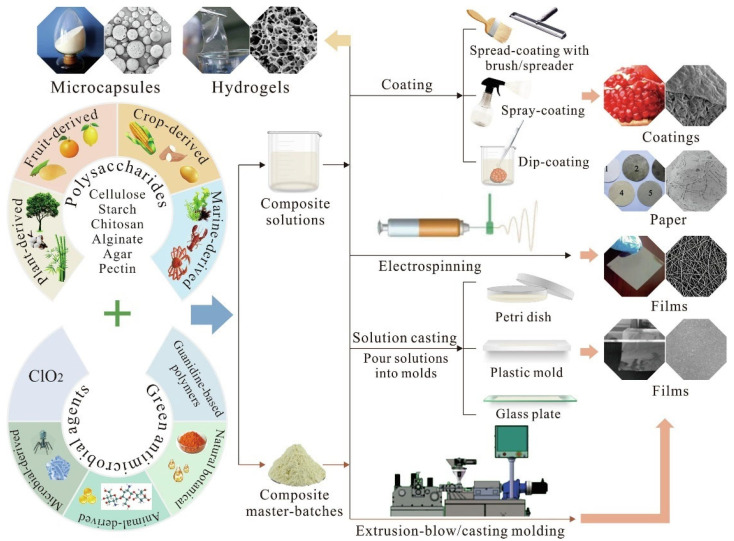
Different applications of green antimicrobial compounds to polysaccharide-based packaging (reproduced with copyright permission from Zhao et al. [107]).

**Figure 6 polymers-14-04257-f006:**
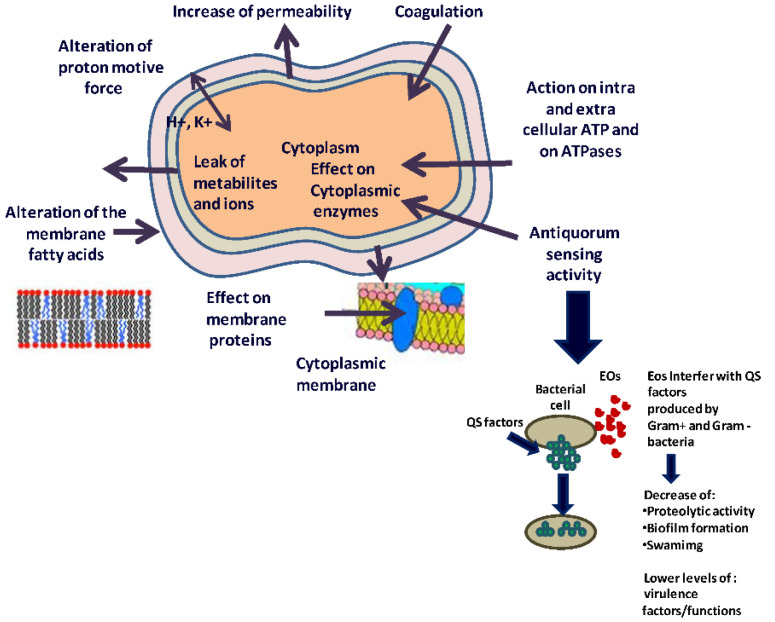
Effect of EOs on cellular membrane and target sites (reproduced with copyright permission from Nazzaro et al. [109]).

**Figure 7 polymers-14-04257-f007:**
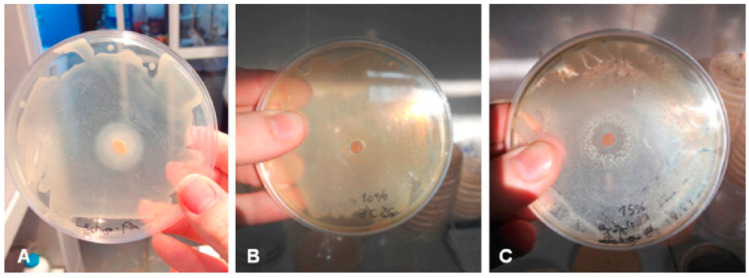
Radial disk diffusion assay on (**A**) lettuce microflora film of control; (**B**) film with 10% thyme essential oil in the presence of *Escherichia coli* and (**C**) film with 15% thyme essential oil against broccoli microflora (reproduced with copyright permission from Chen et al. [24]).

**Figure 8 polymers-14-04257-f008:**
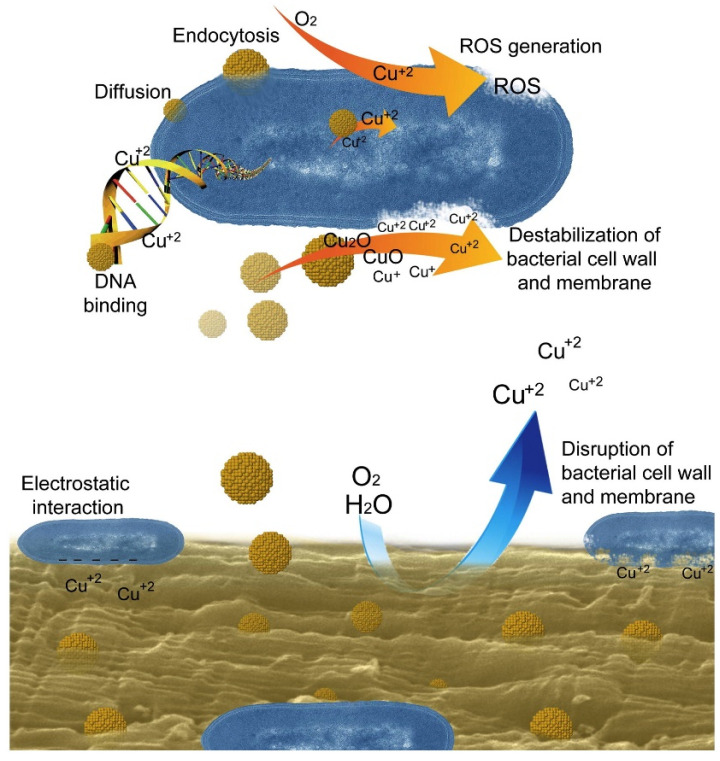
Mechanism of action against bacteria of copper nanoparticles (reproduced with copyright permission from Tamayo et al. [179]).

**Figure 9 polymers-14-04257-f009:**
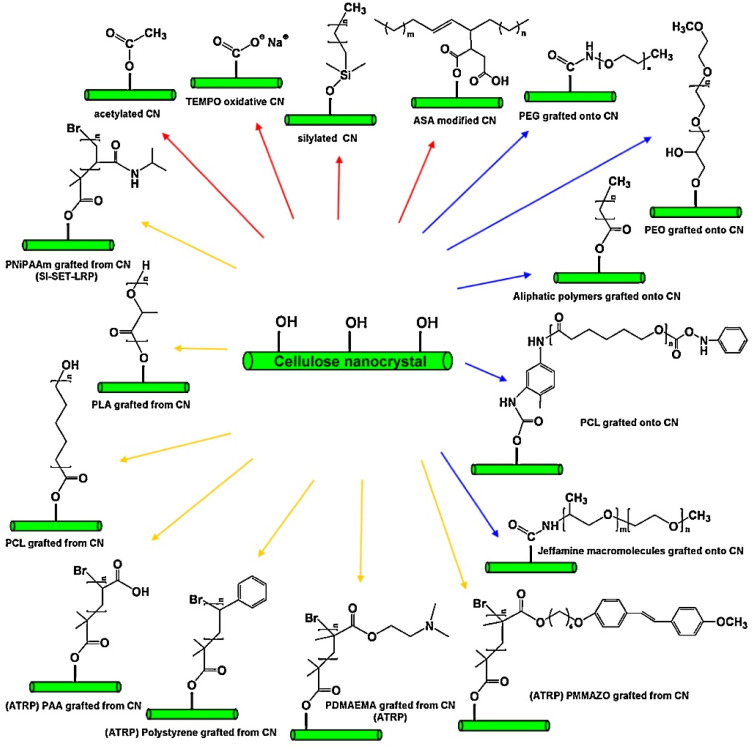
Popular modifications of cellulose nanocrystals (reproduced with copyright permission from Dufresne et al. [196]).

**Figure 10 polymers-14-04257-f010:**
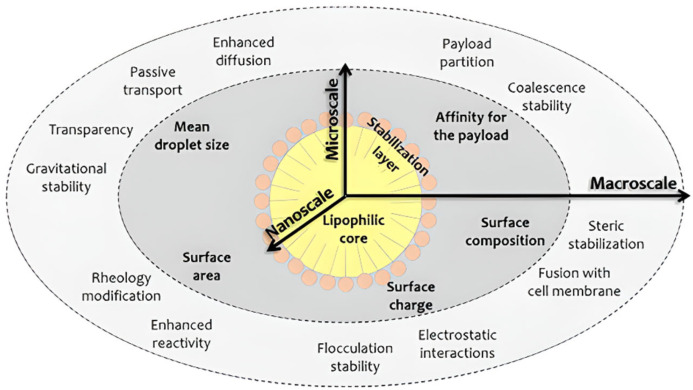
Schematic representation of nanoemulsion and related properties at different sizes (reproduced with copyright permission from Donsì [199]).

**Figure 11 polymers-14-04257-f011:**
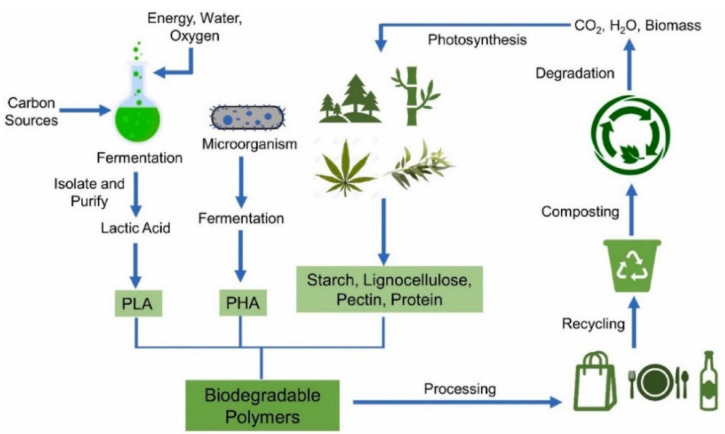
Flow chart for the recycling of polymers in the circular economy (reproduced with copyright permission from Tyagi et al. [57]).

**Table 1 polymers-14-04257-t001:** Cases of study of biopolymers and their effects in food-packaging applications.

Polymers	Additives	Treatments	Solvents for the Polymers	The Effects and Advantages	References
Polybutylene succinate (PBS), Polyhydroxybutyrate (PHB), Polycaprolactone (PCL), Polylactic acid (PLA)	/	Biodegradation test of 10 months at 25, 37, and 50 °C soil and compost	/	Fast degradation of PCL in 8 weeks at 50 °C due to the activity of fungal strain of *T. lanuginosus*	[21]
Poly-β-hydroxybutyrate	/	Fermentation	Oily sludge	Isolated 63 bacterial strains that can produce PHAs Presence of *Bacillus coagulans* in 99.96% of the cases *Bacillus coagulans* showed a production yield with molasses of 6.36 g/L, *B. megaterium*	[25]
PLLA-15% ZIF-8 MOF	/	Extrusion	/	Polymer was not suitable for food packaging because of the high migration level of Zn^2+^ Zn^2+^ release was double in acidic simulant	[26]
Gelatin 6% (*w/v*)	Glutaraldehyde (GTA) 50% (*w/v* of polymer)	Cross-linking	Distilled water	GTA cross-linking enhanced gelatin thermal stability and mechanical properties with pH 4.5	[27]
Methyl cellulose (MC) 1% (*w/v*)	Murta berry extract (MU) 25% (*w/w* of the polymer)Glutaraldehyde (GA) 10–20% (*w/w* of polymer) Polyethylene glycol (PEG) (25% *w/w* of the polymer)	/	Distilled water	Cross-linking decreased the swelling index of the materials and increased mechanical properties	[28]
Binary blend of polymers at a final concentration of 5% of gelatin (GEL) and different polysaccharides: gum arabic (GAR), methylcellulose (MC), octenyl succinic anhydride modified starch (OSA), and water-soluble soy polysaccharides (WSSP)	Glycerol 1% (*w/w*)	/	Distilled water	Pure OSA film had low plasticity GAR film was weak from a mechanical point of view Incompatibility between GEL and MC, especially at a 50/50 ratio GEL improved the durability and stiffness of the film	[29]
Polylactic acid (PLA)	Nanocomposite films containing 1−5% (*w/w* of the polymer) of dye−clay hybrid nano pigments (DCNP)	Cationic exchange reaction between a cationic dye and C20A	Chloroform	Maximum improvement of E′ and glass transition temperature at 3% of DCNP loading level Oxygen permeability and WVP decreased in comparison to neat PLA Optimum of 3% for optical property and UV barrier	[30]
Polylactic acid (PLA) 1% (*w/v*)	α-costic acid (α-CA) 7:1 (*w/w* of the polymer)	/	Chloroform	The plasticising effect of the sesquiterpenoid plant metabolite induced better thermal degradation Homogeneous and efficient inclusion of α-CA in PLA	[31]
PLA latex	Nanocellulose fibrils with high lignin content (NCFHL) from 5 to 20% (*w/w*)	Extraction of *Thuja plicata* bark and fibrillation	Aqueous suspensions of NCFHL	PLA reacted with NCFHL at nearly 50% of the total area NCFHL until 10% enhanced elastic modulus and tensile strength NCFHL increased thermal stability and hydrophobicity	[32]
Fossil-based and bio-based polycarbonate (PC)	/	Moulding	/	Bio-based PC had weak thermal resistance and low viscosity Good optical property but lower birefringence compared with the fossil PC	[7]
Sodium alginate 1–3.5% (*w/v*)	200–800 mg/L of protease from *Bacillus brevis* 1–3.5% CaCl_2_	/	Milli-Q water	∙ Best performance of immobilisation at 2.5–3% of Na-alginate and CaCl_2_, with 400–600 mg/L of protease	[33]
CMC 0.5% (*w/v*), gelatin (GEL) 0.05–0.25% (*w/v*)	Sodium benzoate 5–30% (*w/v*), saturate vapour of glutaraldehyde (GLA)	UV irradiation at (253.7 nm, 30 W) for 30–180 min	Aqueous solution	Best crosslinking rate with 20% SB and 180 min of UV and 0.2% gelatin, associated with exposure of GLA saturate vapour for 90 min Photo-crosslinking enhanced hydrophobicity Both crosslinking methods improved the tensile strength and contact angle of CMC film	[34]
Chitosan 1.5% (*w/v*)	Glycerol 30% (*w/w* of the polymer) and tween 80 0.2% (*w/v* of essential oil)	/	Distilled water	HAE’s gave light barrier properties, higher water content and solubility EOs enhanced tensile strength	[35]
Chitosan (CH) 2% (*w/w*), pea starch (S) 2% (*w/w*), CH:S 1:4 (*w/w*)	Lyophilised tannic acid (TA) 1:0.04 (*w/w* on polymer) or thyme extract (TE) 1:0.15 (*w/w* on polymer)	/	Water dispersion	TE modified the microstructural appearance of CH, due to crosslinking effect of polyphenols TA and TE gave higher resistance at the break but poor elasticity and opaque films	[36]
Corn starch and polylactic acid (PLA) blended at a ratio of 80:20	Epoxidised cardoon oil (ECO) 3% (*w/w* of PLA fraction) and glycerol at 30% (*w/w* of starch fraction)	Melt blending process	/	Compatibility between ECO and PLA, which gave higher WVP and barrier to O_2_ Poor mechanical property of the films	[37]
Zein (Z) 15% (*w/v*), gelatin (G) 10% (*w/v*), blend ZG at different ratios (2:1, 1:1, 1:2) 15% (*w/v*)	Tea polyphenol 2.5%–7.5% (*w/v*), glycerol 0.4–0.8 mL	/	Acetic acid (AA) and water	Zein/gelatin ratio influenced mechanical property in multilayer films Multilayers were more transparent and had a higher UV barrier than neat polymers	[38]
Microcrystalline cellulose 3% (*w/w*)	68% ZnCl_2_ (*w/w*)	/	Distilled water	Transparent Zn-cellulose film crosslinked with Ca^2+^	[39]
Polyvinyl alcohol (PVA) 5–12.5% (*w/v*)	Heat cross-linking with citric acid (CA) 3–12% (*w/w* of the polymer), Clove oil (CO) 20% (*w/w* of the polymer)	Electrospinning and cross-linking	Distilled water	Cross-linking process permitted to reach a swelling degree above 400% Microfibers treated with CA were highly hygroscopic Cross-link improved mechanical property and thermal stability	[40]
Chitosan(CS) 4% (*w/v*)	Whey protein isolate (WPI) 4% (*w/v*), microcrystalline cellulose (MCC) 4% (*w/v*) and glycerin 10–50%	/	Distilled water	Compatibility between polymer and additives Better WVP at 1.5:1 CS/MCC ratio with 30% glycerin and 3.6 of pH	[41]
Gelatin 6% (*w/v*)	*Galla chinensis* extract powder (GCE) 0.03–0.12 g/100 mL	/	Distilled water	GCE worked as a crosslinker for gelatin hydrogel The maximum concentration of GCE improved thermal stability and gel strength	[42]

**Table 2 polymers-14-04257-t002:** Antimicrobial compounds and their efficacy against food-borne pathogenic micro-organisms.

Antimicrobial Compounds	Polymers	Solvents for the Antimicrobial Compounds	The Effects and Advantages	Microorganisms	Efficacy	References
Citric acid 0.5–1% (*w/w*)	Gelatin 2% (*w/v*)	Distilled water	The active coating decreased the microbial charge by 3 logs after 4 days of storage Citric acid helped avoiding lipid oxidation and keeping the pH at values lower than the control	Total bacterial count(TBC)	+	[88]
*L. curvatus* CRL705 bacteriocins	Wheat gluten	Distilled water	The bacteriocins were effective against *L. innocua* but did not affect *L. plantarum* CRL691, probably due to the high concentration of fat in wieners	*Lactobacillus plantarum*	-	[97]
				*Listeria innocua*	+	
Microfluidiser apple skin extract (ASP) 1:1 (*v/v* of the polymer) and tartaric acid (TA) 0.5–1%	0.75% CMC	Distilled water	The ASP/CMC film showed good inhibition zone against *Salmonella enterica* and *Shigella flexneri* regardless of the concentration	*Listeria monocytogenes*	-	[98]
				*Staphylococcus aureus*	-	
				*Salmonella enterica*	+	
				*Shigella flexneri*	+	
AgNPs of 41 and 100 nm	HPMC 3% (*w/w*) in a PVA-coated silver nanoparticles solution	Distilled water	The film of HPMC with nanoparticles showed antibacterial properties against gram-positive *S. aureus* The size of nanoparticles seemed to be influenced by this property	*Escherichia* *coli*	+	[99]
				*Staphylococcus aureus*	+	
Nisin (N), glutaraldehyde (G) and succinic acid (A)	Stainless steel (S)/polydopamine (D)	/	Antimicrobial activity of SDGN and SDAN against *L. monocytogenes*	*Listeria monocytogenes*	+	[100]
Murta berry extract (MU) 25% (*w/w* of the polymer), glutaraldehyde (GA) 10–20% (*w/w* of polymer)	Methyl cellulose (MC) 1% (*w/v*)	Ethanol solution 70%	All the films were effective against *Listeria* In the presence of MU, the reduction percentage of the microorganism was 99.9%	*Listeria innocua*	+	[28]
TiO_2_ nanopowders 0–2% (*w/w*)	Chitosan 2% (*w/v*)	Distilled water	The best results were represented by CS and CT1-UV, due to the intrinsic antimicrobial property of CS and the photocatalysis of TiO_2_ that happens in presence of UV-light	*S. aureus*	+	[101]
				*E. coli*	+	
				*P. aeruginosa*	+	
				*S.* Typhimurium	+	
				*Aspergillus* spp.	+	
				*Pennicillium* spp.	+	
Whey Protein Isolate (WPI)	Clay composite 5–20%	Distilled water	The percentage of clay composite did not influence the antimicrobial effect WPI had a bacteriostatic effect on gram-positive bacteria such as *Listeria monocytogenes*	*Listeria monocytogenes*	+	[102]
				*Escherichia coli*	-	
Nisin (N) 0.25–0.5% (*w/w*) and ε-polylysine (PL) 0.2% (*w/w*)	Corn distarch phosphate 3% (*w/w*),nanocellulose 0.5% (*w/w*), CMC 0.8% (*w/w*)	Distilled water	N showed a better antimicrobial property against *S. aureus*, PL against *E. coli* The combination of the two compounds gave the better result	*S. aureus*	+	[103]
				*E. coli*	+	
Nisin 10^5^ IU/mL in 0.02 M HCl, Grape seed extract 0.5% (*w/v*)	Chitosan 1% (*v/v*), gelatin 3% (*v/v*)	Distilled water	The blend between chitosan and gelatin showed a good antimicrobial property related to the polymers	Total Viable Count (TVC)	+	[104]
Cellulose nanocrystal (CNC) 1% and lignin nanoparticle (LNP) 3%	PLA grafted with GMA at 15% (*w/w* of the polymer)	/	LNP was effective against *P. syringae* pv. *omato* (Pst), even at the concentration of 10^6^ CFU/mL	*Pseudomonas syringae* pv. tomato	+	[105]
Clove oil (CO) 20% (*w/w* of the polymer)	Polyvinyl alcohol (PVA) 5–12.5% (*w/v*) cross-linked with citric acid (CA) 3–12% (*w/w* of the polymer)	Distilled water	CO was particularly effective against *S. aureus,* slowing down the growth of 0.13 OD with respect to the control CO was less effective against *E. coli*	*S. aureus*	+	[40]
				*E. coli*	+	
*Cedrus deodara* pine needle extract (PNE) 15% (*w/w* of SPI) and cellulose nanofibril (CNF) 15% (*w/w* of SPI)	Soy protein isolate (SPI) 6% (*w/v*)	Distilled water	SL film had good activity against all the pathogens tested in this experiment, gram-positive and negative. PNE showed a significant antimicrobial property	*Escherichia coli*	+	[106]
				*Salmonella*Typhimurium	*+*	
				*Staphylococcus* *aureus*	*+*	
				*Listeria* *monocytogenes*	*+*	
Nanosized TiO_2_ 1% (*w/v*) and black plum peel extract (BPPE) 1% (*w/v*)	Chitosan 2% (*w/v*)	Distilled water	Synergistic effect between CS- TiO_2_- BPPE with the highest value of antimicrobial activity All the compounds showed good efficiency against all the tested microorganisms	*Escherichia coli*	+	[75]
				*Staphylococcus aureus*	*+*	
				*Salmonella* spp.	*+*	
				*Listeria monocytogenes*	*+*	

**Table 3 polymers-14-04257-t003:** EOs and their activity through incorporation in packaging.

EOs and Plants Extracts	Polymers	Solvents for EOs	The Effect and Advantages	References
Rosemary EO at 0.5, 1.0, and 1.5% (*v/v*)	Chitosan 2% (*w/v*)	Distilled water	Significant antioxidant activity of EO EO increased WVP and transparency EO reduced UV transmittance	[113]
Extracts of cinnamon, guarana, rosemary and boldo-do-chile	Blend of gelatin 4% and chitosan 1% (*w/v*)	Absolute ethanol	Extracts increased gloss and mechanical property GEL50:CH50 enhanced antioxidant and antimicrobial properties.	[114]
Eugenol (E) and ginger (G) EOs (0.5 g/g biopolymer)	Blend of gelatin 4% and chitosan 1% (*w/v*)	Distilled water	E improved UV-vis light barrier and mechanical properties E showed the greatest resistance to oxidation	[115]
EOs of *Cinnamomum* ssp. and *Syzygium aromaticum*	Chitosan	Ethanol	EOs inhibited more than 95% of mycelial growth of *M. canis* at 200 μg mL^−1^, 100% over 400 μg mL^−1^	[116]
D-Limonene and terpenes from *Melaleuca alternifolia* (25 g/L to 0.1 g/L)	/	Sunflower oil and palm oil	Nanoencapsulated terpenes at 1.0 g/L delayed the microbial growth of *L. delbrueckii*, at 5.0 g/L and completely inactivated the microorganism in fruit juices	[117]
Cinnamon, citronella, pink clove, nutmeg and thyme EOs at 1% (*v/v*)	Chitosan 2%, gelatin 2% (*w/v*)	Distilled water	Compatibility between polymers and EOs EOs improved UV barrier properties Efficient antimicrobial properties for thyme EO against common food pathogens.	[111]
Cinnamon essential oil (EO) 5–15 g/L	Sodium alginate 0.75%, CMC 0.25%	Distilled water	EO gave antimicrobial properties against *E. coli* and *S. aureus* EO enhanced hydrophobicity of the film, thickness, E%, and decreased TS	[55]
Cinnamaldehyde 5.33%	Chitosan 1.5%	Ethanol 96%	High temperature activated the film for the release of the antimicrobial compound that effectively inhibited *L. monocytogenes* in milk	[49]
*Origanum vulgare L.* EO 0.4–1.2%, (*w/v*)	Chitosan nanoparticles (CSNPs) and fish gelatin 4%	Distilled water	EO increased elasticity of the film and ensured a good antimicrobial property against *S. aureus*, *L. monocytogenes*, *S. enteritidis*, and *E. coli* at the concentration of 1.2%	[118]
Oregano EO 0–2% (*v/v*)	Mucilage from quince seeds 1%	Distilled water	Antimicrobial activity against gram-positive bacteria (*S. aureus, L. monocytogenes*) at a concentration higher than 1%	[119]
Carvacrol 0–10% (*w/v* of the polymer)	Cellulose acetate (CA) 5% (*w/v*)	Acetone	10% concentration was effective against gram-positive and gram-negative bacteria and did not change the film characteristics, except for the degree of crystallinity and glass transition temperature CA-carvacrol enhanced three times the shelf life of cooked ham	[120]
Oregano essential-oil nanoemulsion (ORNE) 0–7.5% (*v/v*)	Hydroxypropyl methylcellulose (HPMC) 2.5%	Distilled water	EO at different percentages modulated the mechanical property of the film ORNE improved UV barrier property and showed activity against all the tested microorganisms	[121]
Oregano essential-oil nanoemulsion (ORNE) 0–7.5% (*v/v*)	Fish gelatin 3%, Chitosan 2%	Distilled water	EO gave antioxidant and antimicrobial properties to the film EO positively influenced light barrier and water vapour barrier property (WVP), elasticity and thickness	[122]
Cinnamon EO	Chitosan nanofibre (CSNF) emulsified in Nanostructured lipid carriers (NLC)	Molten cocoa butter	CSNF and EO synergised together, giving a hydrophobic characteristic to the film EO opacified the film	[123]
Clove bud EO 0%–1.5%	Pectin 3% (*w/v*)	Distilled water	EO improved the thermal stability of the film. Efficiency of EO against gram-positive bacteria in agar disc-diffusion assay.	[124]
*Satureja khuzestanica Jamzad* EO 1%	Lecithin:cholesterol (60:0, 50:10, 40:20, and 30:30) dissolved in dichloromethane/methanol (1:1), added to chitosan 2% (*w/v*)	Methanol	Nano-encapsulated EO provided a good extension of the shelf-life of meat lamb products, decreasing the microbial count during storage EO provided antioxidant property	[125]
*T. moroderi* (TM) and *T. piperella* (TP) extracted EOs 0.5–2% (*v/v*)	Chitosan 2% (*w/v*)	Distilled water	High antioxidant activity due to the presence of EOs of plants related to *Thymus* spp., with a higher value for the TP extract Concentrations of 1–2% were effective against all the microorganisms, probably due to the presence of carvacrol or camphor in the EOs, as bioactive compounds	[126]
Clove bud, tagetes, thyme, eucalyptus, neem, cinnamon leaf, himalayan pine needle, tea tree EOs 0–40% (*v/w*)	poly(3-hydroxybutyrate-co-4-hydroxybutyrate) 4%	Chloroform	Thyme oil was the best option among the tested EOs, giving a good antimicrobial property to the film at 30% with the absence of mould EOs acted as a plasticiser for the polymer and increased the WVP and elongation at break (%).	[127]
Plant EOs extracted from *Cinnamomum cassia Presl, Litsea cubeba, Cymbopogon martini, Thymus mongolicus Ronn, Syringa Linn.,* *Lavendula angustifolia Mill., Foeniculum uulgare Mill, Citrus reticulata Banco, Mentha haplocalyx Briq., Allium sativum* and *Artemisia argyi*	/	/	Cinnamomum cassia Presl, Litsea cubeba, Cymbopogon martini and Thymus mongolicus Ronn were the EOs with the best antifungal activities, probably due to the presence of trans-cinnamaldehyde, citral, trans-geraniol, and carvacrol, respectively.	[112]

**Table 4 polymers-14-04257-t004:** Case studies of antioxidant compounds applied to food packaging for the prolongation of the shelf-life.

Antioxidant Compounds	Polymers	Solvents of Antioxidant Compounds	The Effects and Advantages	References
Catechin (2% or 5%) or green tea extract (2% or 5%)	Polypropylene	/	Better stability against thermal oxidation 6 times higher than the control	[153]
Microfluidiser apple skin extract (ASP) 1:1 (*v/v* of the polymer) and tartaric acid (TA) 0.5–1%	0.75% CMC	Distilled water	ASP enhanced the antioxidant activity at every concentration, but 2% was the best one	[98]
Murta berry extract (MU) 25% (*w/w* of the polymer), glutaraldehyde (GA) 10–20% (*w/w* of polymer)	Methyl cellulose (MC) 1% (*w/v*)	Solution in ethanol 70% (*v/v*)	Absence of radical scavenging activity for control without MU GA decreased antioxidant activity at higher concentration MU increased the release of antioxidants from films by up to 50%	[28]
Thyme extract (TE) with a ratio of 0.04:1 on the polymer	Chitosan 2% (*w/w*) and pea starch 2% (*w/w*) solutions blended together in a ratio of 1:4 *w/w*	Ethanol 50%	TE had an antioxidant activity of 0.26 ± 0.02 kg TE/mol DPPH	[36]
Tea polyphenol 2.5%–7.5% *w/w*	Zein (Z) 15% (*w/v*), gelatin (G) 10% (*w/v*), blend ZG at different ratios (2:1, 1:1, 1:2) 15% (*w/v*)	Acetic acid (AA) and water	Tea polyphenol-loaded film inhibited microbial growth and improved water retention on freshly cut fruits.	[38]
*Cedrus deodara* pine needle extract (PNE) 15% (*w/w* of SPI) and cellulose nanofibril (CNF) 15% (*w/w* of SPI)	Soy protein isolate (SPI) 6% (*w/v*)	Distilled water	PNE is rich in phenolic compounds, such as 2R,3R-dihydromyricetin, myricetin-3-O-ß-D-glucopyranoside and protocatechuic acid, that gave antioxidant activity to SLE and SLEC films	[106]
Anthocyanins from black plum peel extract (BPPE) 1% (*w/v*)	Chitosan 2% (*w/v*)Nanosized TiO_2_ 1% (*w/v*)	Distilled water	CS and CS-TiO_2_ showed only a slight antioxidant activity CS-BPPE showed a better radical scavenging activity due to the anthocyanins CS-TiO_2_- BPPE exhibited an intermediate result due to the antagonistic interaction between TiO_2_ and BPPE	[75]

**Table 5 polymers-14-04257-t005:** Application of nanoparticle technologies to food packaging.

Nanoencapsulated or Nanofiller Molecules	Polymers	Solvent	The Effects and Advantages	References
Silver nanoparticles (AgNPs) of 79 mM silver nitrate incapsulated in 45 mM of Poly(vinyl alcohol) (PVA)	Hydroxypropyl methylcellulose (HPMC) 3% (*w/w*)	Distilled water	AgNPs helped to increase tensile strength NPs decreased the WVP	[99]
Montmorillonite clay (MMT) 1–10%	Potato starch (PS) and Microcrystalline cellulose (MCC)	Distilled water	Opacity increased MMT improved thermal stability at higher concentration Compatibility between MMT and PSMCC that increased WVP and mechanical property MMT influenced dielectric property	[172]
TiO_2_ nanopowders 0–2% (*w/w*)	Chitosan 2% (*w/v*)	Distilled water	The addition of nanopowders improved the mechanical and water barrier properties TiO_2_ lowered the transmittance through the film Nanocomposites gave ethylene-photodegradation property to the film	[101]
Nisin (N) 0.25–0.5% (*w/w*) and ε-polylysine (PL) 0.2% (*w/w*)	Corn distarch phosphate 3% (*w/w*),nanocellulose 0.5% (*w/w*), CMC 0.8% (*w/w*)	Distilled water	Good compatibility of N and PL with CN to form a compact and homogeneous film	[103]
Amine functionalised mullite fibres(AMUF) from 0.5 to 10 %wt	Polypropylene-grafted-maleic anhydride (PP-g-MA)	o-xylene	Improved thermal stability Up to 5%, AMUF enhanced the Young’s modulus and gave better crystallisation and less fracture in the structure of PP	[173]
Nanofibril of cellulose 10–40 % *w/w* from wheat straw	Polylatic acid (PLA)	/	1) Solid state shearmilling process (SSSM) permitted to maintain good thermal stability for cellulose but decreased the crystallinity index	[174]
Cellulose nanofibril (CNF) 15% (*w/w* of SPI)	Soy protein isolate (SPI) 6% (*w/v*)	Distilled water	CNFs and PNE gave opacity to the film, improving the barrier to UV-light and preventing photo-oxidation	[106]
Microcrystalline cellulose 3% (*w/w*)	Cellulose 3% in 68% ZnCl_2_ (*w/w*)	Distilled water	Developed a transparent Zn-cellulose film crosslinked with Ca^2+^	[39]
Microcrystalline cellulose (MCC) 4% (*w/v*)	Chitosan (CS) 4% (*w/v*), whey protein isolate (WPI) 4% (*w/v*) and glycerin 10–50%	Distilled water	Compatibility between polymer and additives, rough surface, no sign of pores and cracks Better WVP at 1.5:1 CS/MCC ratio with 30% glycerin and 3.6 of pH	[41]
Nanosized TiO_2_ 1% (*w/v*) and black plum peel extract (BPPE) 1% (*w/v*)	Chitosan 2% (*w/v*)	Distilled water	Nanoencapsulation of TiO_2_ and anthocyanins of BPPE improved mechanical, UV-vis, WVP and light barrier properties Compatibility between molecules that form the film	[75]

**Table 6 polymers-14-04257-t006:** Advantages and disadvantages of essential oils, LABs, biopolymers, nanotechnology, and natural antimicrobials.

Topics	Advantages	Disadvantages	References
Essential oils	Antimicrobial effect Antioxidant property	Volatility Hydrophobicity Modify the flavour of the products	[109,129,224]
LABs	Does not affect the flavour of the products Produce bacteriocins	Different characteristics and effectiveness for every strain Viability	[141,150]
Biopolymers	Biodegradable Renewable Non-toxic Can be extracted from industry agricultural wastes	Water-soluble Poor mechanical property Scarce heat resistance Expensive industrialisation	[3,15,17,18,19,24,45,46,47,51,52,53,54,57,60,63,65,66,67,69,71,72,73,74,76,79,80,81,82,83,84,91,168,199,225,226]
Nanotechnology	Enhancing water vapour property Good barrier to gas Improvement of mechanical property Improvement of thermal stability Wide application in food packaging Antimicrobial and antifungal activity	Decrease of elongation % Potential toxicity Lack of data about migration from packaging to food	[90,167,170,178,179,182,187,188,196,203,227]
NaturalAntimicrobials	Generate ROS to inactivate the bacteria Considered GRAS and non-toxic Biodegradable The one derived from plants usually are good antioxidant compounds Can be encapsulated for a slow release of antimicrobial compounds and a better thermal stability	Poor knowledge of possible interactions between antimicrobial compounds and food Oscillating stability of the effectiveness of these molecules Could give undesirable flavours Only a few natural antimicrobics compounds have a wide range of applications against microorganisms	[1,5,19,20,50,90,91,92,96,107,108,110,133,139,145,147,148,154,179,203,222,228]

## Data Availability

Not applicable.

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
