# Peer review of "The Green Era of Food Packaging: General Considerations and New Trends"

_polymers, 2022, doi:10.3390/polym14204257_

Round 1
Reviewer 1 Report
The topic of this review is very much in line with the current global trends and hot spots in this field, and the summary of research in related fields is also very comprehensive and comprehensive. It is a good summary article, and reviewers feel that they can be hired. However, there are still some problems that suggest the author to consider revision and improvement. I believe it will be more helpful to improve the citation of the article and pay attention to this field, and it will also help the author to have a more comprehensive understanding of the relevant progress in this field. First, the object of the current review is not comprehensive enough, for example, the research object of some of the articles listed below also belongs to this SCOPE, but it has not been reviewed and analyzed, and it is suggested to strengthen it. Second, it is suggested that some methods such as LCA evaluation, especially in the field of plastic packaging, should also be described as a separate chapter. For the application of some related materials in the field of packaging, it is suggested that the author should consider relevant articles such as FPSL and PTS. The reviewers are familiar with the relevant fields, and only enumerate the latest high-level papers in some relevant journals published by scholars from various countries in the past two years for the author's reference. Congratulations on this good article to be published as soon as possible.
Poly(ε-caprolactone): A potential polymer for biodegradable food packaging applicationThe
Active cellulose acetate-carvacrol films: Antibacterial, physical and thermal properties
The future of polyethylene terephthalate bottles: Challenges and sustainability
Life cycle assessment (LCA) of PET and PLA bottles for the packaging of fresh pasteurised milk: The role of the manufacturing process and the disposal scenario
PLLA-ZIF-8 metal organic framework composites for potential use in food applications: Production, characterization and migration studies
PhysicomeThe review can potentially be a great contribution to the special community and well organized in the present form. However, it can be improved to be better as follows.
Poly(ε-caprolactone): A potential polymer for biodegradable food packaging application
The Active cellulose acetate-carvacrol films: Antibacterial, physical and thermal properties
The future of polyethylene terephthalate bottles: Challenges and sustainability
Life cycle assessment (LCA) of PET and PLA bottles for the packaging of fresh pasteurised milk: The role of the manufacturing process and the disposal scenario
PLLA-ZIF-8 metal organic framework composites for potential use in food applications: Production, characterization and migration studies
Physicomechanical, thermal and dielectric properties of eco-friendly starch-microcrystalline cellulose-clay nanocomposite films for food packaging and electrical applications
Fabrication, characterization, and antibacterial properties of citric acid crosslinked PVA electrospun microfibre mats for active food packaging
Preparation and characterization of whey protein isolate/chitosan/microcrystalline cellulose composite films
chanical, thermal and dielectric properties of eco-friendly starch-microcrystalline cellulose-clay nanocomposite films for food packaging and electrical applications
Fabrication, characterization, and antibacterial properties of citric acid crosslinked PVA electrospun microfibre mats for active food packaging
Preparation and characterization of whey protein isolate/chitosan/microcrystalline cellulose composite films
Author Response
We wish to express our deepest gratitude for the careful revision. We found comments and suggestions extremely useful and inspired by the true aim of improving the quality of the paper. We have accepted suggestions and requests for changes, which can be identified with the red color inside the manuscript. Following, response to the comments for manuscript revision considering the reviewer comments:
- It is suggested that some methods such as LCA evaluation, especially in the field of plastic packaging, should also be described as a separate chapter.
LCA chapter has been included in the section “5.1. Life Cycle Assessment LCA”
- The object of the current review is not comprehensive enough, for example, the research object of some of the articles listed below also belongs to this SCOPE, but it has not been reviewed and analyzed, and it is suggested to strengthen it. For the application of some related materials in the field of packaging, it is suggested that the author should consider relevant articles such as FPSL and PTS. The reviewers are familiar with the relevant fields, and only enumerate the latest high-level papers in some relevant journals published by scholars from various countries in the past two years for the author's reference.
All the papers suggested have been added in the tables of the manuscript and cited respectively:
- Poly(ε-caprolactone): A potential polymer for biodegradable food packaging application [83]
- Active cellulose acetate-carvacrol films: Antibacterial, physical and thermal properties [122]
- The future of polyethylene terephthalate bottles: Challenges and sustainability [12]
- Life cycle assessment (LCA) of PET and PLA bottles for the packaging of fresh pasteurised milk: The role of the manufacturing process and the disposal scenario [214]
- PLLA-ZIF-8 metal organic framework composites for potential use in food applications: Production, characterization and migration studies [26]
- Physicomechanical, thermal and dielectric properties of eco-friendly starch-microcrystalline cellulose-clay nanocomposite films for food packaging and electrical applications [175]
- Fabrication, characterization, and antibacterial properties of citric acid crosslinked PVA electrospun microfibre mats for active food packaging [40]
- Preparation and characterization of whey protein isolate/chitosan/microcrystalline cellulose composite films [41]

Reviewer 2 Report
The paper presents a review about biodegradable packaging material and its application as an alternative packaging material in green packaging trend. In this paper, the author categorized all kinds of biodegradable packaing materials in detail. Also, the author analysis the availability of the material properties in packaging fields. Then the paper introduced upon antibacterial material and its application in packaging field. Finally, the author give conclusion about biodegradable packaing, providing some suggestions in biogradable future development. However, there are still some points worth discussing.My detailed comments are as follows:
1. Abstract should be written in a logic way without dividing it intodiferent points. Please refer the abstract of other published papers.
2. Adjust the sharpness of the referenced images, or download the images from the journal's website to keep the images with high resolution and good readability.
3. The paper presented biopolymer in details and divided into several categories. To fully demonstrated different biopolymer varies particular biogradable features, the author should add information on degradation method, degradatioin rate and its residual, etc. 4. As a review article, comprehensive article colloction is necessary. Please carefully read, discuss and cite the following articles regarding the packaging: Packaging and degradability properties of polyvinyl alcohol/gelatin nanocomposite films filled water hyacinth cellulose nanocrystals; Development and characterization of food packaging bioplastic film from cocoa pod husk cellulose incorporated with sugarcane bagasse fibre; etc. 5. Figure 5 is not that clear in categories criterion. Generally speaking, antibacteria packaging on the basis of perparation methods can devied into antibacterial composite material, blending antibacterial material, antibacterial coating and antibacterial polymer. Authors are suggested to provide a better classifications in this figure.6. Cellulose is an essential biodegradable material. Hence, the author should add relevant preparaption methods, structural, physical and chemical properties about CNC, CNF, etc.
7. Currently, one of the big challenge in biogradable packaging field is hard to produce in large scale with low cost. Thus, in future challenge part, the author should add difficulties in this issue.
8. Some recent highly relevent review articles on the packaging materials should be refered in the manuscript: Nanomaterials 2020, 10(1), 150; https://doi.org/10.3390/nano10010150; Nanomaterials 2022, 12(18), 3158; https://doi.org/10.3390/nano12183158; etc.
9. There are still some formatting errors and syntax errors that need to be carefully corrected.
10. Please carefully check the reference list to ensure the full information is provided, such as page numbers.
Author Response
We wish to express our deepest gratitude for the careful revision. We found comments and suggestions extremely useful and inspired by the true aim of improving the quality of the paper. We have accepted suggestions and requests for changes, which can be identified with the red color inside the manuscript. Following, response to the comments for manuscript revision considering the reviewer comments:
My detailed comments are as follows:
- Abstract should be written in a logic way without dividing it into different points. Please refer the abstract of other published papers.
Done.
- Adjust the sharpness of the referenced images, or download the images from the journal's website to keep the images with high resolution and good readability.
All the images are now updated with a minimum of 1000x1000 pixels as required by polymers instructions. In addition, under every figure, the following sentence have been inserted: “reproduced with copyright permission from Author et al. [citation]”.
- The paper presented biopolymer in details and divided into several categories. To fully demonstrated different biopolymer varies particular biogradable features, the author should add information on degradation method, degradatioin rate and its residual, etc.
Information and new citations about the degradation of every biopolymer have been added in the relative section.
- As a review article, comprehensive article colloction is necessary. Please carefully read, discuss and cite the following articles regarding the packaging: Packaging and degradability properties of polyvinyl alcohol/gelatin nanocomposite films filled water hyacinth cellulose nanocrystals; Development and characterization of food packaging bioplastic film from cocoa pod husk cellulose incorporated with sugarcane bagasse fibre; etc.
Thank you for the interesting papers, we read, discussed and cited them with [201] and [218].
- Figure 5 is not that clear in categories criterion. Generally speaking, antibacteria packaging on the basis of perparation methods can devied into antibacterial composite material, blending antibacterial material, antibacterial coating and antibacterial polymer. Authors are suggested to provide a better classification in this figure.
Due to the absence in the literature of a sufficiently explanatory image regarding the five classes of antimicrobials with permission of the copyright, figure 5 has been replaced with another one that explain the methods of application and the different ways of including components.
- Cellulose is an essential biodegradable material. Hence, the author should add relevant preparaption methods, structural, physical and chemical properties about CNC, CNF, etc.
More papers have been studied and cited for an extensive exploration and clarification of CNCs, CNFs and BNCs, with particular attention to methods of extractions. This part has been added in the “4.1.4. Bio-nanofillers” section of the manuscript.
- Currently, one of the big challenge in biogradable packaging field is hard to produce in large scale with low cost. Thus, in future challenge part, the author should add difficulties in this issue.
The issue is now listed in the section “6. Future challenges and concluding remarks” as one of the main gaps between laboratory scale and commercialization. The problem has been discussed with some consideration.
- Some recent highly relevent review articles on the packaging materials should be refered in the manuscript: Nanomaterials 2020, 10(1), 150; https://doi.org/10.3390/nano10010150; Nanomaterials 2022, 12(18), 3158; https://doi.org/10.3390/nano12183158; etc.
Like the 4° point, we studied and cited the papers [187, 194] which have been included in the section “4.1.4. Bio-nanofillers” of the manuscript.
- There are still some formatting errors and syntax errors that need to be carefully corrected
Done.
- Please carefully check the reference list to ensure the full information is provided, such as page numbers.
Done.

Round 2
Reviewer 2 Report
Accept in present form